# Multi-ancestry GWAS of the electrocardiographic PR interval identifies 202 loci underlying cardiac conduction

Ioanna Ntalla et al.[#]

The electrocardiographic PR interval reflects atrioventricular conduction, and is associated with conduction abnormalities, pacemaker implantation, atrial fibrillation (AF), and cardiovascular mortality. Here we report a multi-ancestry ($N = 293,051$) genome-wide association meta-analysis for the PR interval, discovering 202 loci of which 141 have not previously been reported. Variants at identified loci increase the percentage of heritability explained, from 33.5% to 62.6%. We observe enrichment for cardiac muscle developmental/contractile and cytoskeletal genes, highlighting key regulation processes for atrioventricular conduction. Additionally, 8 loci not previously reported harbor genes underlying inherited arrhythmic syndromes and/or cardiomyopathies suggesting a role for these genes in cardiovascular pathology in the general population. We show that polygenic predisposition to PR interval duration is an endophenotype for cardiovascular disease, including distal conduction disease, AF, and atrioventricular pre-excitation. These findings advance our understanding of the polygenic basis of cardiac conduction, and the genetic relationship between PR interval duration and cardiovascular disease.

[#]A full list of authors and their affiliations appears at the end of the paper.

The electrocardiogram is among the most common clinical tests ordered to assess cardiac abnormalities. Reproducible waveforms indicating discrete electrophysiologic processes were described over 100 years ago, yet the biological underpinnings of conduction and repolarization remain incompletely defined. The electrocardiographic PR interval reflects conduction from the atria to ventricles, across specialized conduction tissues such as the atrioventricular node and the His-Purkinje system. Pathological variation in the PR interval may indicate heart block or pre-excitation, both of which can lead to sudden death[1]. The PR interval also serves as a risk factor for atrial fibrillation and cardiovascular mortality[1–3]. Prior genetic association studies have identified 64 PR interval loci[4–13]. Yet the underlying biological mechanisms of atrioventricular conduction and relationships between genetic predisposition to PR interval duration and disease are incompletely characterized.

To enhance our understanding of the genetic and biological mechanisms of atrioventricular conduction, we perform genome-wide association studies (GWAS) meta-analyses of autosomal and X chromosome variants mainly imputed with the 1000 Genomes Project reference panel (http://www.internationalgenome.org)[14] of PR interval duration. We then conduct downstream in silico analyses to elucidate candidate genes and key pathways, and examine relationships between genetic variants linked to PR interval duration and cardiovascular disease in the UK biobank (UKB; https://www.ukbiobank.ac.uk). Over 200 loci are genome-wide significant, and our results imply key regulation processes for atrioventricular conduction, and candidate genes in cardiac muscle development/contraction and the cytoskeleton. We observe associations between polygenic predisposition to PR interval duration with distal conduction disease, AF, and atrioventricular pre-excitation. Our findings highlight the polygenic basis of atrioventricular conduction, and the genetic relationship between PR interval duration and other cardiovascular diseases.

## Results

**Meta-analysis of GWASs**. We performed a primary meta-analysis including 293,051 individuals of European (92.6%), African (2.7%), Hispanic (4%), and Brazilian (<1%) ancestries from 40 studies (Supplementary Data 1 and 2, Supplementary Table 1). We also performed ancestry-specific meta-analyses (Fig. 1). A total of 202 genome-wide significant loci ($P < 5 \times 10^{-8}$) were identified in the multi-ancestry analysis, of which 141 were not previously reported (Supplementary Data 3, Fig. 2, Supplementary Figs. 1 and 2). We considered for discovery only variants present in >60% of the maximum sample size in the GWAS summary results, a filtering criterion used to ensure robustness of associated loci (median proportion of sample size included in analyses for lead variants 1.0, interquartile range 0.99–1.00; Methods). There was strong support in our data for all 64 previously reported loci (61 at $P < 5 \times 10^{-8}$ and 3 at $P < 1.1 \times 10^{-4}$; Supplementary Data 4 and 5). In a secondary analysis among the European ancestry subset, a total of 127 loci not previously reported reached genome-wide significance (Supplementary Data 6, Supplementary Figs. 1–4), of which lead variants at 8 loci were borderline genome-wide significant ($P < 9.1 \times 10^{-7}$) in our multi-ancestry meta-analysis. None of the previously unreported loci were genome-wide significant in African or Hispanic/Latino ancestry meta-analyses (Supplementary Data 7, Supplementary Figs. 1 and 3). We observed no genome-wide significant loci in the X chromosome meta-analyses (Supplementary Fig. 5). In sensitivity analyses, we examined the rank-based inverse normal transformed residuals of PR interval. Results of absolute and transformed trait meta-analyses were highly correlated ($P > 0.94$, Supplementary Data 8–10, Supplementary Figs. 6 and 7).

By applying joint and conditional analyses in the European meta-analysis data, we identified multiple independently associated variants ($P_{\text{joint}} < 5 \times 10^{-8}$ and $r^2 < 0.1$) at 12 previously not reported and 25 previously reported loci (Supplementary Data 11). The overall variant-based heritability ($h^2_g$) for the PR interval estimated in 59,097 unrelated European participants from the UKB with electrocardiograms was 18.2% (Methods). In the UKB, the proportion of $h^2_g$ explained by variation at all loci discovered in our analysis was 62.6%, compared with 33.5% when considering previously reported loci only.

We annotated variants at 149 loci (141 previously not reported loci from the multi-ancestry meta-analysis and 8 loci from the meta-analysis of European ancestry subset). The majority of the lead variants at the 149 loci were common (minor allele frequency, MAF > 5%). We observed 6 low-frequency (MAF 1–5%) variants, and one rare (MAF < 1%) predicted damaging missense variant. The rare variant (rs35816944, p.Ser171Leu) is in *SPSB3* encoding SplA/Ryanodine Receptor Domain and SOCS Box-containing 3. SPSB3 is involved in degradation of the transcription factor SNAIL, which regulates the epithelial-mesenchymal transition[15], and has not been previously associated with cardiovascular traits. At *MYH6*, a previously described locus for PR interval[6,10], sick sinus syndrome[16], AF and other cardiovascular traits[17], we observed a previously not reported predicted damaging missense variant in *MYH6* (rs28711516, p. Gly56Arg). *MYH6* encodes the α-heavy chain subunit of cardiac myosin. In total, we identified missense variants in genes at 11 previously not reported loci, one from the European subset meta-analysis, and 6 previously reported loci (Supplementary Data 12). These variants are a representation of multiple variants at each locus, which are in high LD, and thus may not be the causative variant.

**Expression quantitative trait loci (eQTLs)**. PR interval lead variants (or best proxy [$r^2 > 0.8$]) at 43 previously not reported and 23 previously reported loci were significant cis-eQTLs (at a 5% false discovery rate (FDR) in left ventricle (LV) and right atrial appendage (RAA) tissue samples from the Genotype-Tissue Expression (GTEx; https://gtexportal.org/home/) project[18]. Variants at 13 previously not reported and 6 previously reported loci were eQTLs in spleen, which was used as negative control tissue (Supplementary Data 13). The PR interval associations and eQTLs colocalized at 31 previously not reported loci and 14 previously reported loci (posterior probability [PP] > 75%. Variants at 9 previously not reported loci were significant eQTLs only in LV and RAA tissues with consistent directionality of gene expression.

**Predicted gene expression**. In an exploratory analysis, we also performed a transcriptome-wide analysis to evaluate associations between predicted gene expression in LV and RAA with the PR interval. We identified 113 genes meeting our significance threshold ($P < 3.1 \times 10^{-6}$, after Bonferroni correction), of which 91 were localized at PR interval loci (within 500 kb from a lead variant; Supplementary Data 14, Supplementary Fig. 8). Longer PR interval duration was associated with decreased levels of predicted gene expression for 57 genes, and increased levels for 56 genes (Fig. 3). In spleen tissues, only 31 gene expression-PR interval associations were detected, and 19 of them did not overlap with the findings in heart tissues.

**Regulatory annotation of loci**. Most PR interval variants were annotated as non-coding. Therefore, we explored whether associated variants or proxies were located in transcriptionally active genomic regions. We observed enrichment for DNase

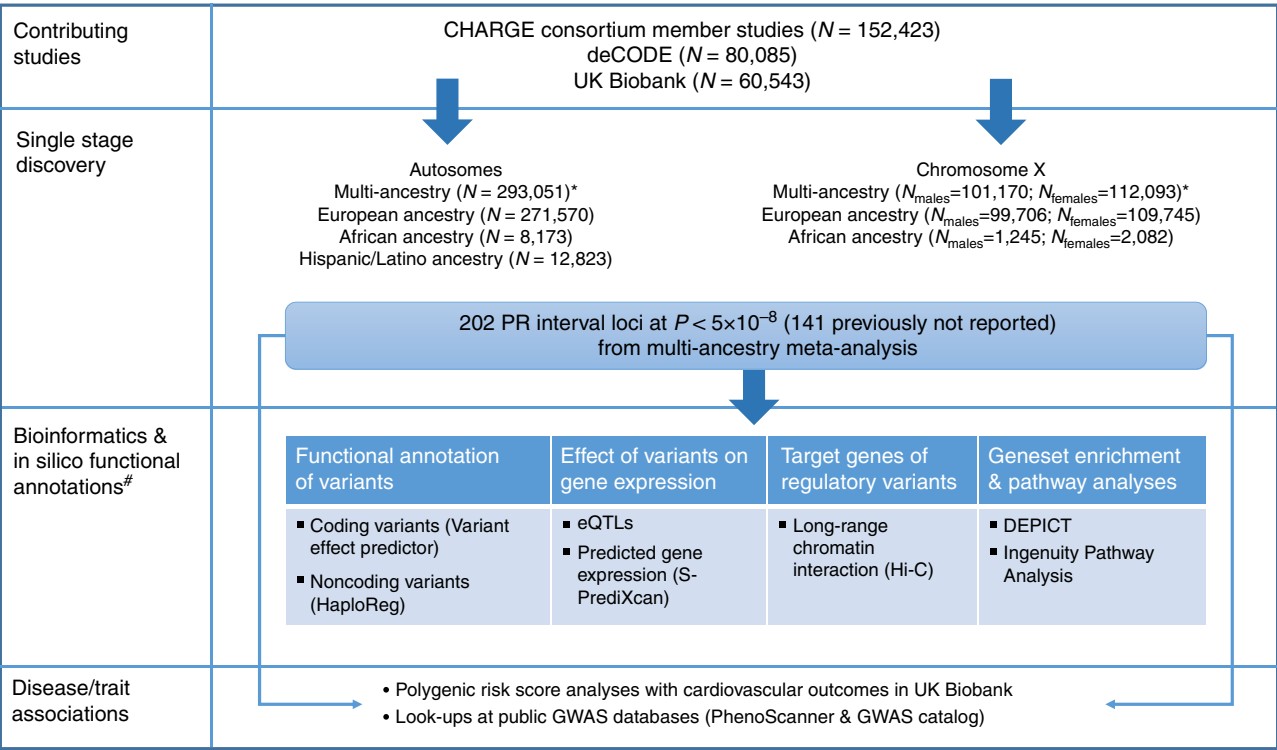

**Fig. 1 Overview of the study design.** An overview of contributing studies, single-stage discovery approach, and downstream bioinformatics and in silico annotations performed to link variants to genes, and polygenic risk score analysis to link variants to cardiovascular disease risk is illustrated. Asterisk (*) The multi-ancestry meta-analysis is our primary analysis. Previously not reported loci were identified from the multi-ancestry meta-analysis. Ancestry specific and chromosome X meta-analysis are secondary. Hash (#) For bioinformatics and in silico annotations we also included loci that reached genome-wide significance in European only meta-analysis ($N = 8$) and were borderline genome-wide significant in the multi-ancestry meta-analysis.

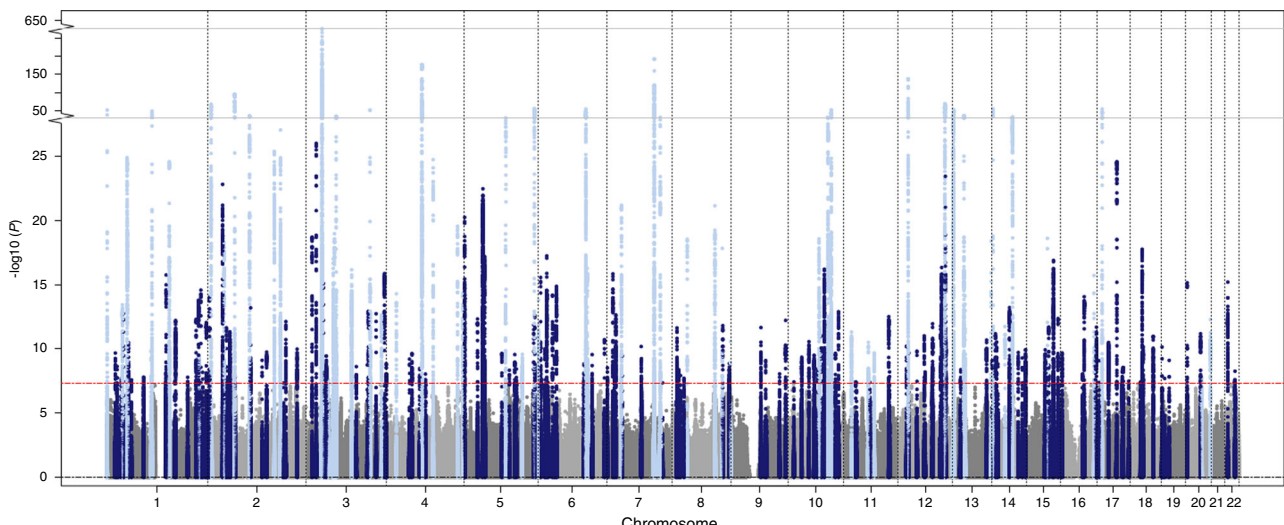

**Fig. 2 Manhattan plot of the multi-ancestry meta-analysis for PR interval.** P values are plotted on the -log$_{10}$ scale for all variants present in at least 60% of the maximum sample size from the fixed-effects meta-analysis of 293,051 individuals from multiple ancestries (multi-ancestry meta-analysis). Associations of genome-wide significant ($P < 5 \times 10^{-8}$) variants at previously not reported ($N = 141$) and previously reported loci ($N = 61$) are plotted in dark and light blue colors respectively.

I-hypersensitive sites in fetal heart tissue ($P < 9.36 \times 10^{-5}$, Supplementary Fig. 9). Analysis of chromatin states indicated variants at 97 previously not reported, 6 European, and 52 previously reported loci were located within regulatory elements that are present in heart tissues (Supplementary Data 15), providing support for gene regulatory mechanisms in specifying the PR interval. To identify

distal candidate genes at PR interval loci, we assessed the same set of variants for chromatin interactions in a LV tissue Hi-C dataset[19]. Forty-eight target genes were identified (Supplementary Data 16). Variants at 35 previously not reported and 3 European loci were associated with other traits, including AF and coronary heart disease (Supplementary Data 17, Supplementary Fig. 10).

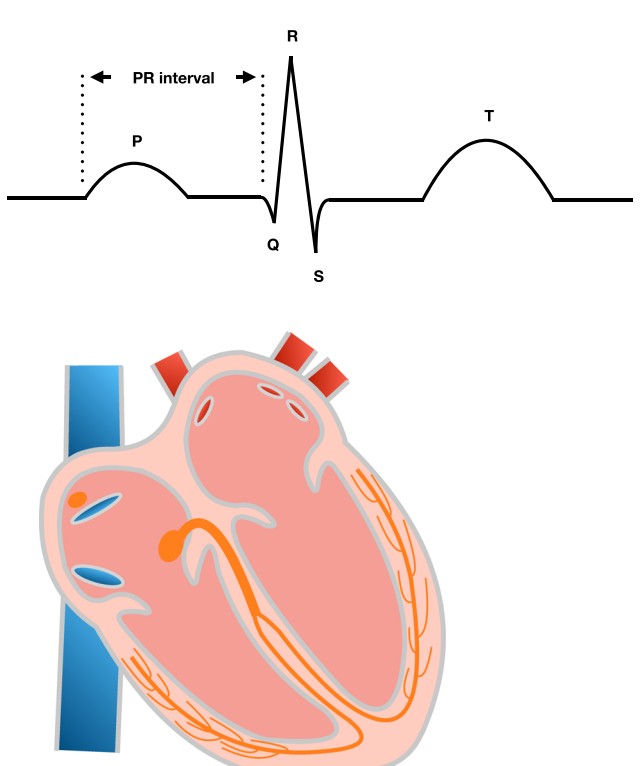

## Gene expression related to longer PR

| | | | |
|---|---|---|---|
| ACP6 | DNM1P51 | MYO15A | TCTN3 |
| AL590822.1 | EDN2 | NPIPA5 | TMEM182 |
| ALPK3 | EFNA1 | NUDT13 | TPMT |
| ATP5D | FADS1 | PDZRN3 | TRAK1 |
| BMPR1A | FAM211B | PHACTR1 | TRIP4 |
| C11orf1 | FAT1 | RP11-29H23.5 | TTC18 |
| CALHM2 | FKBP7 | RP11-399K21.11 | VDAC2 |
| CAMK2D | FUT11 | RP11-3B7.1 | VPREB3 |
| CCDC36 | GBAP1 | RP4-764O22.2 | XIRP1 |
| CDH13 | HMGA1P5 | RPSA | ZCCHC24 |
| CEFIP | IFRD2 | SLC25A26 | ZNF503-AS1 |
| CFDP1 | KCND3 | SLC6A6 | |
| CHRM2 | KDM1B | SLK | |
| DAG1 | LRCH1 | SNX1 | |
| DEK | MSTO2P | SYNPO2L | |

## Gene expression related to shorter PR

| | | | |
|---|---|---|---|
| AC011747.4 | IL17D | PLCD1 | SPATA20 |
| AC103965.1 | IL25 | PPAPDC3 | SPTBN1 |
| AGAP5 | KPNA3 | QRICH1 | SSBP3 |
| BEND7 | LINC00964 | RCAN2 | SSXP10 |
| C1orf86 | MALAT1 | RP11-1070N10.3 | STRN |
| CAB39L | MLF1 | RP11-182J1.16 | SYNE2 |
| CBX8 | MMP11 | RP11-344N10.5 | SYPL2 |
| CMTM5 | MRPL37 | RP11-379F4.7 | TFEC |
| CSPG4P11 | MTSS1 | RP11-397E7.4 | THRB |
| DDX42 | MYBPHL | RP11-724N1.1 | UBE3B |
| DNAH11 | MYOZ1 | SCN5A | WDR73 |
| EMB | NDST2 | SCN10A | ZHX1 |
| GBF1 | NEURL | SH3PXD2A | |
| GORASP1 | NPIPA1 | SLC2A11 | |
| HCN1 | PHLDB2 | SMARCB1 | |

**Fig. 3 Plausible candidate genes of PR interval from S-PrediXcan.** Diagram of standard electrocardiographic intervals and the heart. The electrocardiographic features are illustratively aligned with the corresponding cardiac conduction system structures (orange) reflected on the tracing. The PR interval (labeled) indicates conduction through the atria, atrioventricular node, His bundle, and Purkinje fibers. Right: Supplementary Data 14 shows 113 genes whose expression in the left ventricle ($N = 233$) or right atrial appendage ($N = 231$) was associated with PR interval duration in a transcriptome-wide analysis using S-PrediXcan and GTEx v7. Displayed genes include those with significant associations after Bonferroni correction for all tested genes ($P < 3.1 \times 10^{-6}$). Longer PR intervals were associated with increased predicted expression of 56 genes (blue) and reduced expression of 57 genes (orange).

**In silico functional annotation and pathway analysis.** Bioinformatics and in silico functional annotations for potential candidate genes at the 149 loci are summarized in Supplementary Data 18 and 19. Using a prior GWAS of AF[20,21], we identified variants with shared associations between PR interval duration and AF risk (Supplementary Fig. 11). Enrichment analysis of genes at PR interval loci using Data driven Expression-Prioritized Integration for Complex Traits (DEPICT: https://data.broadinstitute.org/mpg/depict/)[22] indicated heart development ($P = 1.87 \times 10^{-15}$) and actin cytoskeleton organization ($P = 2.20 \times 10^{-15}$) as the most significantly enriched processes (Supplementary Data 20 and 21). Ingenuity Pathway Analysis (IPA; https://www.qiagenbioinformatics.com/products/ingenuity-pathway-analysis/) supported heart development, ion channel signaling and cell-junction/cell-signaling amongst the most significant canonical pathways (Supplementary Data 22).

**Polygenic risk scores (PRSs) with cardiovascular traits.** Finally, we evaluated associations between genetic predisposition to PR interval duration and 16 cardiac phenotypes chosen a priori using ~309,000 unrelated UKB European participants not included in our meta-analyses[23]. We created a PRS for PR interval using the European ancestry meta-analysis results (Fig. 4, Supplementary Table 2). Genetically determined PR interval prolongation was associated with higher risk of distal conduction disease (atrioventricular block; odds ratio [OR] per standard deviation 1.11, $P = 7.02 \times 10^{-8}$) and pacemaker implantation (OR 1.06, $P = 1.5 \times 10^{-4}$). In contrast, genetically determined PR interval prolongation was associated with reduced risk of AF (OR 0.95, $P = 4.30 \times 10^{-8}$) and was marginally associated with a reduced risk of

atrioventricular pre-excitation (Wolff–Parkinson–White syndrome; OR 0.85, $P = 0.003$). Results were similar when using a PRS derived using the multi-ancestry meta-analysis results (Supplementary Fig. 12, Supplementary Table 2, and Supplementary Data 3).

## Discussion

In a meta-analysis of nearly 300,000 individuals, we identified 202 loci, of which 141 were previously not reported underlying cardiac conduction as manifested by the electrocardiographic PR interval. Apart from confirming well-established associations in loci harboring ion-channel genes, our findings further underscore the central importance of heart development and cytoskeletal components in atrioventricular conduction[10,12,13]. We also highlight the role of common variation at loci harboring genes underlying monogenic forms of arrhythmias and cardiomyopathies in cardiac conduction.

We report signals in/near 12 candidate genes at previously not reported loci with functional roles in cytoskeletal assembly (*DSP, DES, OBSL1, PDLIM5, LDB3, FHL2, CEFIP, SSPN, TLN2, PTK2, GJA5,* and *CDH2*; Fig. 5). *DSP* and *DES* encode components of the cardiac desmosome, a complex involved in ionic communication between cardiomyocytes and maintenance of cellular integrity. Mutations in the desmosome are implicated in arrhythmogenic cardiomyopathy (ACM) and dilated cardiomyopathy (DCM)[24–28]. Conduction slowing is a major component of the pathophysiology of arrhythmia in ACM and other cardiomyopathies[29,30]. *OBSL1* encodes obscurin-like 1, which together with obscurin (OBSCN) is involved in sarcomerogenesis by bridging titin (TTN) and myomesin at the M-band[31]. *PDLIM5*

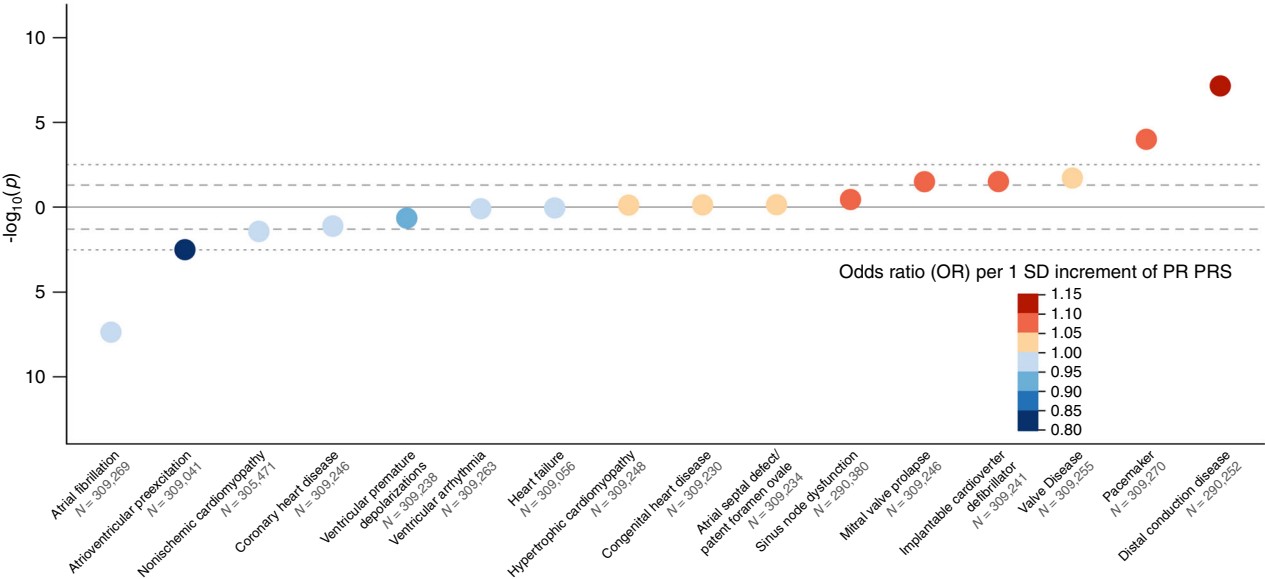

**Fig. 4 Bubble plot of phenome-wide association analysis of European ancestry PR interval polygenic risk score.** The polygenic risk score was derived from the European ancestry meta-analysis. Orange circles indicate that polygenic predisposition to longer PR interval is associated with an increased risk of the condition, whereas blue circles indicate that polygenic predisposition to longer PR interval is associated with lower risk of the condition. The darkness of the color reflects the effect size (odds ratio, OR) per 1 standard deviation (s.d.) increment of the polygenic risk score from logistic regression. Sample size (N) in each regression model is provided under X-axis. Given correlation between traits, we set significance threshold at $P < 3.13 \times 10^{-3}$ after Bonferroni correction ($P < 0.05/16$; dotted line) for the analysis and also report nominal associations ($P < 0.05$; dashed line).

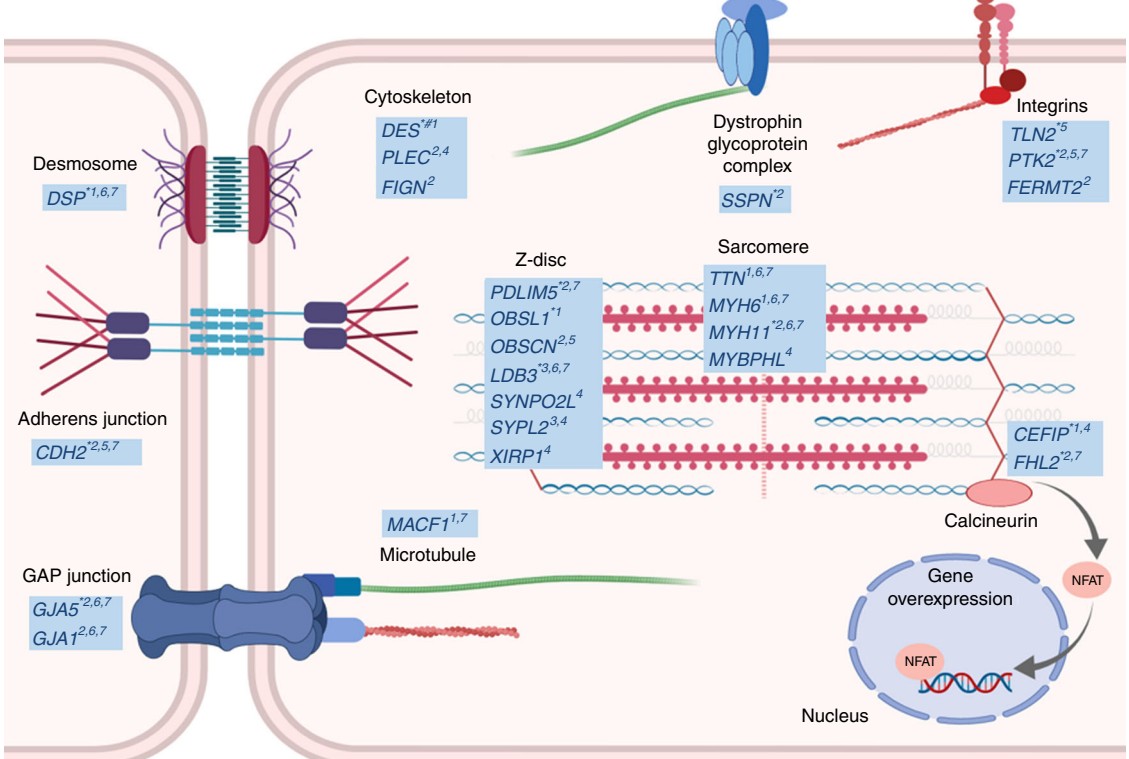

**Fig. 5 Candidate genes in PR interval loci encoding proteins involved in cardiac muscle cytoskeleton.** Candidate genes or encoded proteins are indicated by a star symbol in the figure and are listed in Supplementary Data 3. More information about the genes is provided in Supplementary Data 18 and 19. This figure was created with BioRender. *Previously not reported locus, # genome-wide significant locus in transformed trait meta-analysis. [1]Missense variant; [2]Nearest gene to the lead variant; [3]Gene within the region ($r^2 > 0.5$); [4]Variant(s) in the locus are associated with gene expression in left ventricle and/or right atrial appendage; [5]Left ventricle best HiC locus interactor (RegulomeDB score ≤ 2); [6]Animal model; [7]Monogenic disease with a cardiovascular phenotype.

encodes a scaffold protein that tethers protein kinases to the Z-disk, and has been associated with DCM in homozygous murine cardiac knockouts[32]. *FHL2* encodes calcineurin-binding protein four and a half LIM domains 2, which is involved in cardiac development by negatively regulating calcineurin/NFAT signaling in cardiomyocytes[33]. Missense mutations in *FHL2* have been associated with hypertrophic cardiomyopathy[34]. *CEFIP* encodes the cardiac-enriched FHL2-interacting protein located at the Z-disc, which interacts with *FHL2*. It is also involved in calcineurin–NFAT signaling, but its overexpression leads to cardiomyocyte hypertrophy[35].

Common variants in/near genes associated with monogenic arrhythmia syndromes were also observed, suggesting these genes may also affect atrioventricular conduction and cardiovascular pathology in the general population. Apart from *DSP, DES,* and *GJA5* discussed above, our analyses indicate 2 additional candidate genes (*HCN4* and *RYR2*). *HCN4* encodes a component of the hyperpolarization-activated cyclic nucleotide-gated potassium channel which specifies the sinoatrial pacemaker "funny" current, and is implicated in sinus node dysfunction, AF, and left ventricular noncompaction[36–38]. *RYR2* encodes a calcium channel component in the cardiac sarcoplasmic reticulum and is implicated in catecholaminergic polymorphic ventricular tachycardia[39].

Genes with roles in autonomic signaling in the heart (*CHRM2, ADCY5*) were indicated from expression analyses (Supplementary Data 13 and 18). *CHRM2* encodes the M2 muscarinic cholinergic receptors that bind acetylcholine and are expressed in the heart[40]. Their stimulation results in inhibition of adenylate cyclase encoded by *ADCY5*, which in turn inhibits ion channel function. Ultimately, the signaling cascade can result in reduced levels of the pacemaker "funny" current in the sinoatrial and atrioventricular nodes, reduced L-type calcium current in all myocyte populations, and increased inwardly rectifying $I_{K.Ach}$ potassium current in the conduction tissues and atria causing cardiomyocyte hyperpolarization[41]. Stimulation has also been reported to shorten atrial action potential duration and thereby facilitate re-entry, which may lead to AF[42–44].

By constructing PRSs, we also observed that genetically determined PR interval duration is an endophenotype for several adult-onset complex cardiovascular diseases, the most significant of which are arrhythmias and conduction disorders. For example, our findings are consistent with previous epidemiologic data supporting a U-shaped relationship between PR interval duration and AF risk[2]. Although aggregate genetic predisposition to PR interval prolongation is associated with reduced AF risk, top PR interval prolonging alleles are associated with decreased AF risk (e.g., localized to the *SCN5A/SCN10A* locus; Supplementary Fig. 11) whereas others are associated with increased AF risk (e.g., localized to the *TTN* locus; Supplementary Fig. 11), consistent with prior reports[8]. These findings suggest that genetic determinants of the PR interval may identify distinct pathophysiologic mechanisms leading to AF, perhaps via specifying differences in tissue excitability, conduction velocity, or refractoriness. Future efforts are warranted to better understand the relations between genetically determined PR interval and specific arrhythmia mechanisms.

In conclusion, our study more than triples the reported number of PR interval loci, which collectively explain ~62% of trait-related heritability. Our findings highlight important biological processes underlying atrioventricular conduction, which include both ion channel function, and specification of cytoskeletal components. Our study also indicates that common variation in Mendelian cardiovascular disease genes contributes to population-based variation in the PR interval. Lastly, we observe that genetic determinants of the PR interval provide novel insights into the etiology of several complex cardiac diseases,

including AF. Collectively, our results represent a major advance in understanding the polygenic nature of cardiac conduction, and the genetic relationship between PR interval duration and arrhythmias.

## Methods

**Contributing studies.** A total of 40 studies (Supplementary Methods) comprising 293,051 individuals of European ($N = 271,570$), African ($N = 8,173$), Hispanic ($N = 11,686$), and Brazilian ($N = 485$) ancestries contributed GWAS summary statistics for PR interval. Study-specific design, sample quality control and descriptive statistics are provided in Supplementary Tables 1–3. For the majority of the studies imputation was performed for autosomal chromosomes and X chromosome using the 1000 Genomes (1000 G: http://www.internationalgenome.org) project[14] reference panel. A few studies used whole genome sequence data and the Haplotype Reference Consortium (HRC: http://www.haplotype-reference-consortium.org)/UK10K and 1000 G phase 3 panel was used for UK Biobank (Full details are provided in Supplementary Table 2).

**Ethical approval.** All contributing studies had study-specific ethical approvals and written informed consent. The details are provided in Supplementary Note 1.

**PR interval phenotype and exclusions.** The PR interval was measured in milliseconds (ms) from standard 12-lead electrocardiograms (ECGs), except in the UK Biobank where it was obtained from 4-lead ECGs (CAM-USB 6.5, Cardiosoft v6.51) recorded during a 15 second rest period prior to an exercise test (Supplementary Methods). We requested exclusion of individuals with extreme PR interval values (<80 ms or >320 ms), second/third degree heart block, AF on the ECG, or a history of myocardial infarction or heart failure, Wolff–Parkinson–White syndrome, those who had a pacemaker, individuals receiving class I and class III antiarrhythmic medications, digoxin, and pregnancy. Where data were available these exclusions were applied.

**Study-level association analyses.** We regressed the absolute PR interval on each genotype dosage using multiple linear regression with an additive genetic effect and adjusted for age, sex, height, body mass index, heart rate and any other study-specific covariates. To account for relatedness, linear mixed effects models were used for family studies. To account for population structure, analyses were also adjusted for principal components of ancestry derived from genotyped variants after excluding related individuals. Analyses of autosomal variants were conducted separately for each ancestry group. X chromosome analyses were performed separately for males and females. Analyses using rank-based inverse normal transformed residuals of PR interval corrected for the aforementioned covariates were also conducted. Residuals were calculated separately by ancestral group for autosomal variants, and separately for males and females for X chromosome variants.

**Centralized quality control.** We performed quality control centrally for each result file using EasyQC version 11.4 (https://www.uni-regensburg.de/medizin/epidemiologie-praeventivmedizin/genetische-epidemiologie/software/#)[45]. We removed variants that were monomorphic, had a minor allele count (MAC) < 6, imputation quality metric <0.3 (imputed by MACH; http://csg.sph.umich.edu/abecasis/mach/tour/imputation.html) or 0.4 (imputed by IMPUTE2; http://mathgen.stats.ox.ac.uk/impute/impute_v2.html), had invalid or mismatched alleles, were duplicated, or if they were allele frequency outliers (difference > 0.2 from the allele frequency in 1000 G project). We inspected PZ plots, effect allele frequency plots, effect size distributions, QQ plots, and compared effect sizes in each study to effect sizes from prior reports for established PR interval loci to identify genotype and study-level anomalies. Variants with effective MAC ( $= 2 \times N \times MAF \times$ imputation quality metric) <10 were omitted from each study prior to meta-analysis.

**Meta-analyses.** We aggregated summary-level associations between genotypes and absolute PR interval from all individuals ($N = 293,051$), and only from Europeans ($N = 271,570$), African Americans ($N = 8,173$), and Hispanic/Latinos ($N = 12,823$) using a fixed-effects meta-analysis approach implemented in METAL (http://csg.sph.umich.edu/abecasis/metal/, release on 2011/03/25)[46]. We considered as primary our multi-ancestry meta-analysis, and ancestry-specific meta-analyses as secondary. For the X chromosome, meta-analyses were conducted in a sex-stratified fashion. Genomic control was applied (if inflation factor $\lambda_{GC} > 1$) at the study level. Quantile–quantile (QQ) plots of observed versus expected $-\log_{10}(P)$ did not show substantive inflation (Supplementary Figs. 1 and 2).

Given the large sample size we undertook a one-stage discovery study design. To ensure the robustness of this approach we considered for discovery only variants reaching genome-wide significance ($P < 5 \times 10^{-8}$) present in at least 60% of the maximum sample size ($N_{max}$) in our GWAS summary results. We denote loci as previously not reported if the variants map outside 64 previously reported loci (Supplementary Methods, Supplementary Data 4) for both the multi-ancestry

and ancestry-specific meta-analysis (secondary meta-analyses). Genome-wide significant variants were grouped into independent loci based on both distance (±500 kb) and linkage disequilibrium (LD, $r^2 < 0.1$) (Supplementary Methods). We assessed heterogeneity in allelic effect sizes among studies contributing to the meta-analysis and among ancestral groups by the $I^2$ inconsistency index[47] for the lead variant in each previously not reported locus. LocusZoom (http://locuszoom.org/)[48] was used to create region plots of identified loci. For reporting, we only declare as previously not reported genome-wide significant loci from our primary meta-analysis. However, we considered ancestry-specific loci for annotation and downstream analyses. The results from secondary analyses are specifically indicated in Supplementary Data 6 and 7.

Meta-analyses (multi-ancestry [$N = 282,128$], European only [$N = 271,570$], and African [$N = 8,173$]) of rank-based inverse normal transformed residuals of PR interval were also performed (sensitivity meta-analyses). Because not all studies contributed summary-level association statistics of the transformed PR interval, we considered as primary the multi-ancestry meta-analysis of absolute PR interval for which we achieved the maximum sample size. Loci that met our significance criteria in the meta-analyses of transformed PR interval were not taken forward for downstream analyses.

**Conditional and heritability analysis.** Conditional and joint GWAS analyses were implemented in GCTA v1.91.3 (https://cnsgenomics.com/software/gcta/#Overview)[49] using summary-level variant statistics from the European ancestry meta-analysis to identify independent association signals within PR interval loci. We used 59,097 unrelated (kinship coefficient >0.0884) UK Biobank participants of European ancestry as the reference sample to model patterns of LD between variants. We declared as conditionally independent any genome-wide significant variants in conditional analysis ($P_{joint} < 5 \times 10^{-8}$) not in LD ($r^2 < 0.1$) with the lead variant in the locus.

Using the same set of individuals from UK Biobank, we estimated the aggregate genetic contributions to PR interval with restricted maximum likelihood as implemented in BOLT-REML v2.3.4 (https://data.broadinstitute.org/alkesgroup/BOLT-LMM/)[50]. We calculated the additive overall variant-heritability ($h^2_g$) based on 333,167 LD-pruned genotyped variants, as well as the $h^2_g$ of variants at PR interval associated loci only. Loci windows were based on both distance (±500 kb) and LD ($r^2 > 0.1$) around previously not reported and reported variants (Supplementary Methods). We then calculated the proportion of total $h^2_g$ explained at PR interval loci by dividing the $h^2_g$ estimate of PR interval loci by the total $h^2_g$.

**Bioinformatics and in silico functional analyses.** We use Variant Effect Predictor (VEP; https://www.ensembl.org/info/docs/tools/vep/index.html)[51] to obtain functional characterization of variants including consequence, information on nearest genes and, where applicable, amino acid substitution and functional impact, based on SIFT[52] and PolyPhen-2[53] prediction tools. For non-coding variants, we assessed overlap with DNase I–hypersensitive sites (DHS) and chromatin states as determined by Roadmap Epigenomics Project[54] across all tissues and in cardiac tissues (E083, fetal heart; E095, LV; E104, right atrium; E105, right ventricle) using HaploReg v4.1 (https://pubs.broadinstitute.org/mammals/haploreg/haploreg.php)[55] and using FORGE (https://github.com/iandunham/Forge).

We assessed whether any PR interval variants were related to cardiac gene expression using GTEx (https://gtexportal.org/home/)[18] version 7 cis-eQTL LV ($N = 233$) and RAA ($N = 231$) European data. If the variant at a locus was not available in GTEx, we used proxy variants ($r^2 > 0.8$). We then evaluated the effects of predicted gene expression levels on PR interval duration using S-PrediXcan (https://github.com/hakyimlab/MetaXcan)[56]. GTEx[18] genotypes (variants with MAF > 0.01) and normalized expression data in LV and RAA provided by the software developers were used as the training datasets for the prediction models. The prediction models between each gene-tissue pair were performed by Elastic-Net, and only significant models for prediction were included in the analysis, where significance was determined if nested cross validated correlation between predicted and actual levels were greater than 0.10 (equivalent to $R^2 > 0.01$) and $P$ value of the correlation test was less than 0.05. We used the European meta-analysis summary-level results (variants with at least 60% of maximum sample size) as the study dataset and then performed the S-PrediXcan calculator to estimate the expression-PR interval associations. For both eQTL and S-PrediXcan assessments, we additionally included spleen tissue in Europeans ($N = 119$) as a negative control. In total, we tested 5366, 5977, and 4598 associations in LV, RAA, and spleen, respectively. Significance threshold of S-PrediXcan was set at $P = 3.1 \times 10^{-6}$ ($= 0.05/(5977 + 5366 + 4598)$) to account for multiple testing. In order to determine whether the GWAS identified loci were colocalized with the eQTL analysis, we performed genetic colocalization analysis for eQTL and S-PrediXcan identified gene regions, using the Bayesian approach in COLOC package (R version 3.5; https://cran.r-project.org/web/packages/coloc/index.html). Variants located within the same identified gene regions were included. We set the significant threshold for the PP (two significant associations sharing a common causal variant) at >75%.

We applied GARFIELD (GWAS analysis of regulatory or functional information enrichment with LD correction; https://www.ebi.ac.uk/birney-srv/GARFIELD/)[57] to analyze the enrichment patterns for functional annotations of

the European meta-analysis summary statistics, using regulatory maps from the Encyclopedia of DNA Elements (ENCODE)[58] and Roadmap Epigenomics[54] projects. This method calculates odds ratios and enrichment $P$ values at different GWAS $P$ value thresholds (denoted T) for each annotation by using a logistic regression model accounting for LD, matched genotyping variants and local gene density with the application of logistic regression to derive statistical significance. Threshold for significant enrichment was set to $P = 9.36 \times 10^{-5}$ (after multiple-testing correction for the number of effective annotations).

We identified potential target genes of regulatory variants using long-range chromatin interaction (Hi-C) data from the LV[19]. Hi-C data was corrected for genomic biases and distance using the Hi-C Pro and Fit-Hi-C pipelines according to Schmitt et al. (40 kb resolution – correction applied to interactions with 50 kb–5 Mb span). We identified the promoter interactions for all potential regulatory variants in LD ($r^2 > 0.8$) with our lead and conditionally independent PR interval variants and report the interactors with the variants with the highest regulatory potential a Regulome DB score of ≤2 (RegulomeDB; http://www.regulomedb.org) to annotate the loci.

We performed a literature review, and queried the Online Mendelian Inheritance in Man (OMIM; https://www.omim.org/) and the International Mouse Phenotyping Consortium (https://www.mousephenotype.org/) databases for all genes in regions defined by $r^2 > 0.5$ from the lead variant at each previously not reported locus. We further expanded the gene listing with any genes that were indicated by gene expression or chromatin interaction analyses. We performed look-ups for each lead variant or their proxies ($r^2 > 0.8$) for associations ($P < 5 \times 10^{-8}$) for common traits using both GWAS catalog[59] and PhenoScanner v2[60] databases. For AF, we summarized the results of lead PR interval variants for PR interval and their associations with AF risk from two recently published GWASs[20,21]. We included variants in high linkage disequilibrium with lead PR variants ($r^2 > 0.7$).

**Geneset enrichment and pathway analyses.** We used DEPICT (https://data.broadinstitute.org/mpg/depict/)[22] to identify enriched pathways and tissues/cell types where genes from associated loci are highly expressed using all genome-wide significant ($P < 5 \times 10^{-8}$) variants in our multi-ancestry meta-analysis present in at least 60% of $N_{max}$ ($N_{variants} = 20,076$). To identify uncorrelated variants for PR interval, DEPICT performed LD-clumping ($r^2 = 0.1$, window size = 250 kb) using LD estimates between variants from the 1000 G reference data on individuals from all ancestries after excluding the major histocompatibility complex region on chromosome 6. Geneset enrichment analysis was conducted based on 14,461 predefined reconstituted gene sets from various databases and data types, including Gene ontology, Kyoto encyclopedia of genes and genomes (KEGG), REACTOME, phenotypic gene sets derived from the Mouse genetics initiative, and protein molecular pathways derived from protein–protein interaction. Finally, tissue and cell type enrichment analyses were performed based on expression information in any of the 209 Medical Subject Heading (MeSH) annotations for the 37,427 human Affymetrix HGU133a2.0 platform microarray probes.

IPA (https://www.qiagenbioinformatics.com/products/ingenuity-pathway-analysis/) was conducted using an extended list comprising 593 genes located in regions defined by $r^2 > 0.5$ with the lead or conditionally independent variants for all PR interval loci, or the nearest gene. We further expanded this list by adding genes indicated by gene expression analyses. Only molecules and/or relationships for human or mouse or rat and experimentally verified results were considered. The significance $P$ value associated with enrichment of functional processes is calculated using the right-tailed Fisher's exact test by considering the number of query molecules that participate in that function and the total number of molecules that are known to be associated with that function in the IPA.

**Associations between genetically determined PR interval and cardiovascular conditions.** We examined associations between genetic determinants of atrioventricular conduction and candidate cardiovascular diseases in unrelated individuals of European ancestry from UK Biobank ($N\sim309,000$ not included in our GWAS meta-analyses) by creating PRSs for PR interval based on our GWAS results. We derived two PRSs. One was derived from the European ancestry meta-analysis results, and the other from the multi-ancestry meta-analysis results. We used the LD-clumping feature in PLINK v1.90[61] ($r^2 = 0.1$, window size = 250 kb, $P = 5 \times 10^{-8}$) to select variants for each PRS. Referent LD structure was based on 1000 G European only, and all ancestry data. In total, we selected 582 and 743 variants from European only and multi-ancestry meta-analysis results, respectively. We calculated the PRSs for PR interval by summing the dosage of PR interval prolonging alleles weighted by the corresponding effect size from the meta-analysis results. A total of 581 variants for the PRS derived from European results and 743 variants for the PRS derived from multi-ancestry results (among the variants with imputation quality >0.6) were included in our PRS calculations.

We selected candidate cardiovascular conditions a priori, which included various cardiac conduction and structural traits such as bradyarrhythmia, AF, atrioventricular pre-excitation, heart failure, cardiomyopathy, and congenital heart disease. We ascertained disease status based on data from baseline interviews, hospital diagnosis codes (ICD-9 and ICD-10), cause of death codes (ICD-10), and operation codes. Details of individual selections and disease definitions are described in Supplementary Data 23.

We tested the PRSs for association with cardiovascular conditions using logistic regression. We adjusted for enrolled age, sex, genotyping array, and phenotype-related principal components of ancestry. Given correlation between traits, we set significance threshold at $P < 3.13 \times 10^{-3}$ after Bonferroni correction ($P < 0.05/16$) for the number of analyses performed and also report nominal associations ($P < 0.05$).

**Reporting summary**. Further information on research design is available in the Nature Research Reporting Summary linked to this article.

## Data availability

Summary GWAS statistics are publicly available on the Cardiovascular Disease Knowledge portal (http://www.broadcvdi.org). All other data are contained in the article file and its supplementary information or is available upon request.

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

## Acknowledgements

We provide all investigator and study-specific acknowledgements in Supplementary Note 1, and funding sources in Supplementary Note 2.

## Author contributions

Interpreted results, writing, and editing the manuscript: I.N., L.-C.W., S.A.L., and P.B.M. Conceptualization and supervision of project: S.A.L. and P.B.M. Contributed to GWAS analysis plan: I.N., L.-C.W., H.R.W., Y.J., S.A.L., and P.B.M. Performed meta-analyses: I.N. and L.-C.W. Performed GCTA, heritability, geneset enrichment, and pathway analyses, variant annotations: I.N. Performed PRS and gene expression analyses: S.H.C., M.D.C., and L.-C.W. Performed HiC analyses: I.N., M.R.B., B.M., and P.B.M. Performed gene literature review: I.N., L.-C.W., A.W.Hall, N.R.T., M.D.C., J.H.C., J.J.C., A.T., Y.J., S.A.L., and P.B.M. Contributed to study-specific GWAS by providing phenotype, gen-otype and performing data analyses: J.M., I.R., C.H., P.G., M.P.C., T.B., O.P., I.K., E.T., N.M.A., R.P.S., M.F.L., A.L.P.R., A.M., V.G., E.I., A.P.M., F.D.M., L.F., M.G., A.A.H., J.P. C., L.L., C.M.L., J.S., N.J.S., C.P.N., M.B.R., S.U., G.S., P.P.M., M.K., N.M., K.N., I.N., M.J. C., A.D., S.P., M.E.M., J.R.O., A.R.S., K.R., D.C., L.R., S.A., S.T., T.L., O.T.R., N.H., L.P.L, J.F.W., P.K.J., C.L.K.B., H.C., C.M.v., J.A.K., A.I., P.L.H., L.-C.W., S.A.L., P.T.E., T.B.H., L. J.L., A.V.S., V.G., E.P.B., R.J.F.L., G.N.N., M.H.P., A.C., H.M., J.W., M.M.-N., A.P., T.M., M.W., T.D.S., Y.J., M.M., M.R., Y.J.V., P.H., N.V., K.S., S.K., K.S., M.F.S., B.L., C.R., D.F., M.J.C., M.O., D.M.R., M.B.S., J.G.S, J.A.B., M.L.B., J.C.B., B.M.P., N.S., K.R., C.P., P.P.P., A.G., C.F., J.W.J., I.F., P.W.M., S.T., S.W., M.D., S.B.F., U.V., A.S.H., A.J., K.S., V.S., S.R. H., J.I.R., X.G., H.J.L., J.Y., K.D.T., R.N., R.d., D.O.M., A.C.M., F.C., J.D., E.G.L., Y.Q., K.V.T., E.J.B., D.L., H.L., C.H.N., K.L.L., A.D.M., D.J.P., B.H.S., B.H.S., M.E.v, A.U., J.H.,

R.D.J., U.P., A.P.R., E.A.W., C.K., E.B., D.E.A., G.B.E., A.A., E.Z.S., C.L.A., S.M.G., K.F.K., C.C.L., A.A.S., A.S., S.A., M.A.S., M.Y.v., P.D.L., A.T., M.O., J.R., S.V.D., P.B.M., K.S., H.H., P.S., G.S., G.T., R.B.T., U.T., D.O.A., D.F.G. All authors read, revised, and approved the manuscript.

## Conflict of interest

I.N. became a full-time employee of Gilead Sciences Ltd following submission of the manuscript. S.A.L. receives sponsored research support from Bristol Myers Squibb/Pfizer, Bayer AG, and Boehringer Ingelheim, and has consulted for Bristol Myers Squibb/Pfizer and Bayer AG. P.T.E. is supported by a grant from Bayer AG to the Broad Institute focused on the genetics and therapeutics of cardiovascular diseases. P. T.E. has also served on advisory boards or consulted for Bayer AG, Quest Diagnostics, and Novartis. M.J.C. is Chief Scientist for Genomics England, a UK Government company. B.M.P. serves on the DSMB of a clinical trial funded by Zoll LifeCor and on the Steering Committee of the Yale Open Data Access Project funded by Johnson & Johnson. V.S. has participated in a conference trip sponsored by Novo Nordisk and received a modest honorarium for participating in an advisory board meeting. K.S., H.H., P.S., G.S., G.T., R.B.T., U.T., D.O.A., D.F.G. are employed by deCODE genetics/ Amgen Inc. E.I. is employed by GlaxoSmithKline. A.M. is employed by Genentech Inc.

## Additional information

Ioanna Ntalla[1,145], Lu-Chen Weng[2,3,145], James H. Cartwright[1], Amelia Weber Hall[2,3], Gardar Sveinbjornsson[4], Nathan R. Tucker[2,3], Seung Hoan Choi[3], Mark D. Chaffin[3], Carolina Roselli[3,5], Michael R. Barnes[1,6], Borbala Mifsud[1,7], Helen R. Warren[1,6], Caroline Hayward[8], Jonathan Marten[8], James J. Cranley[1], Maria Pina Concas[9], Paolo Gasparini[9,10], Thibaud Boutin[8], Ivana Kolcic[11], Ozren Polasek[11,12,13], Igor Rudan[14], Nathalia M. Araujo[15], Maria Fernanda Lima-Costa[16], Antonio Luiz P. Ribeiro[17], Renan P. Souza[15], Eduardo Tarazona-Santos[15], Vilmantas Giedraitis[18], Erik Ingelsson[19,20,21,22], Anubha Mahajan[23], Andrew P. Morris[23,24,25], Fabiola Del Greco M[26], Luisa Foco[26], Martin Gögele[26], Andrew A. Hicks[26], James P. Cook[24], Lars Lind[27], Cecilia M. Lindgren[28,29,30], Johan Sundström[31], Christopher P. Nelson[32,33], Muhammad B. Riaz[32,33], Nilesh J. Samani[32,33],

Gianfranco Sinagra[34], Sheila Ulivi[9], Mika Kähönen[35,36], Pashupati P. Mishra[37,38], Nina Mononen[37,38], Kjell Nikus[39,40], Mark J. Caulfield[1,6], Anna Dominiczak[41], Sandosh Padmanabhan[41,42], May E. Montasser[43,44], Jeff R. O'Connell[43,44], Kathleen Ryan[43,44], Alan R. Shuldiner[43,44], Stefanie Aeschbacher[45], David Conen[45,46], Lorenz Risch[47,48,49], Sébastien Thériault[46,50], Nina Hutri-Kähönen[51,52], Terho Lehtimäki[37,38], Leo-Pekka Lyytikäinen[37,38,39], Olli T. Raitakari[53,54,55], Catriona L. K. Barnes[14], Harry Campbell[14], Peter K. Joshi[14], James F. Wilson[8,14], Aaron Isaacs[56], Jan A. Kors[57], Cornelia M. van Duijn[58], Paul L. Huang[2], Vilmundur Gudnason[59,60], Tamara B. Harris[61], Lenore J. Launer[61], Albert V. Smith[59,62], Erwin P. Bottinger[63], Ruth J. F. Loos[63,64], Girish N. Nadkarni[63], Michael H. Preuss[63], Adolfo Correa[65], Hao Mei[66], James Wilson[67], Thomas Meitinger[68,69,70], Martina Müller-Nurasyid[68,71,72,73], Annette Peters[68,74,75], Melanie Waldenberger[68,75,76], Massimo Mangino[77,78], Timothy D. Spector[77], Michiel Rienstra[5], Yordi J. van de Vegte[5], Pim van der Harst[5], Niek Verweij[5,79], Stefan Kääb[68,73], Katharina Schramm[68,71,73], Moritz F. Sinner[68,73], Konstantin Strauch[71,72], Michael J. Cutler[80], Diane Fatkin[81,82,83], Barry London[84], Morten Olesen[85,86], Dan M. Roden[87], M. Benjamin Shoemaker[88], J. Gustav Smith[89], Mary L. Biggs[90,91], Joshua C. Bis[90], Jennifer A. Brody[90], Bruce M. Psaty[90,92,93], Kenneth Rice[91], Nona Sotoodehnia[90,92,94], Alessandro De Grandi[26], Christian Fuchsberger[26], Cristian Pattaro[26], Peter P. Pramstaller[26], Ian Ford[95], J. Wouter Jukema[96,97], Peter W. Macfarlane[98], Stella Trompet[99], Marcus Dörr[100,101], Stephan B. Felix[100,101], Uwe Völker[100,102], Stefan Weiss[100,102], Aki S. Havulinna[103,104], Antti Jula[103], Katri Sääksjärvi[103], Veikko Salomaa[103], Xiuqing Guo[105], Susan R. Heckbert[106], Henry J. Lin[105], Jerome I. Rotter[105], Kent D. Taylor[105], Jie Yao[107], Renée de Mutsert[108], Arie C. Maan[96], Dennis O. Mook-Kanamori[108,109], Raymond Noordam[99], Francesco Cucca[110], Jun Ding[111], Edward G. Lakatta[112], Yong Qian[111], Kirill V. Tarasov[112], Daniel Levy[113,114], Honghuang Lin[114,115], Christopher H. Newton-Cheh[3,116], Kathryn L. Lunetta[114,117], Alison D. Murray[118], David J. Porteous[119,120], Blair H. Smith[121], Bruno H. Stricker[122], André Uitterlinden[123], Marten E. van den Berg[122], Jeffrey Haessler[124], Rebecca D. Jackson[125], Charles Kooperberg[124], Ulrike Peters[124], Alexander P. Reiner[124,126], Eric A. Whitsel[127], Alvaro Alonso[128], Dan E. Arking[129], Eric Boerwinkle[130], Georg B. Ehret[131], Elsayed Z. Soliman[132], Christy L. Avery[133,134], Stephanie M. Gogarten[91], Kathleen F. Kerr[91], Cathy C. Laurie[91], Amanda A. Seyerle[135], Adrienne Stilp[91], Solmaz Assa[5], M. Abdullah Said[5], M. Yldau van der Ende[5], Pier D. Lambiase[136,137], Michele Orini[136,138], Julia Ramirez[1,137], Stefan Van Duijvenboden[1,137], David O. Arnar[4,60,139], Daniel F. Gudbjartsson[4,140], Hilma Holm[4], Patrick Sulem[4], Gudmar Thorleifsson[4], Rosa B. Thorolfsdottir[4,60], Unnur Thorsteinsdottir[4,60], Emelia J. Benjamin[114,141,142], Andrew Tinker[1,6], Kari Stefansson[4,60], Patrick T. Ellinor[2,3,143], Yalda Jamshidi[144], Steven A. Lubitz[2,3,143,146 ✉] & Patricia B. Munroe[1,6,146 ✉]

¹William Harvey Research Institute, Barts and The London School of Medicine and Dentistry, Queen Mary University of London, London, UK. ²Cardiovascular Research Center, Massachusetts General Hospital, Boston, MA, USA. ³Program in Medical and Population Genetics, The Broad Institute of MIT and Harvard, Cambridge, MA, USA. ⁴deCODE genetics/Amgen, Inc., Reykjavik, Iceland. ⁵Department of Cardiology, University of Groningen, University Medical Center Groningen, Groningen, The Netherlands. ⁶National Institute for Health Research, Barts Cardiovascular Biomedical Research Centre, Queen Mary University of London, London, UK. ⁷College of Health and Life Sciences, Hamad Bin Khalifa University, Education City, Doha, Qatar. ⁸Medical Research Council Human Genetics Unit, Institute of Genetics and Molecular Medicine, University of Edinburgh, Edinburgh, UK. ⁹Institute for Maternal and Child Health-IRCCS 'Burlo Garofolo', Trieste, Italy. ¹⁰Department of Medicine, Surgery and Health Science, University of Trieste, Trieste, Italy. ¹¹University of Split School of Medicine, Split, Croatia. ¹²Clinical Hospital Centre Split, Split, Croatia. ¹³Psychiatric Hospital Sveti Ivan, Zagreb, Croatia. ¹⁴Centre for Global Health Research, Usher Institute of Population Health Sciences and Informatics, University of Edinburgh, Edinburgh, UK. ¹⁵Departamento de Biologia Geral, Universidade Federal de Minas Gerais, Belo Horizonte, Minas Gerais, Brazil. ¹⁶Rene Rachou Reserch Institute, Oswaldo Cruz Foundation, Belo Horizonte, Minas Gerais, Brazil. ¹⁷Hospital das Clínicas e Faculdade de Medicina, Universidade Federal de Minas Gerais, Belo Horizonte, Minas Gerais, Brazil. ¹⁸Department of Public Health, Geriatrics, Uppsala University, Uppsala, Sweden. ¹⁹Department of Medicine, Division of Cardiovascular Medicine, Stanford University School of Medicine, Stanford, CA, USA. ²⁰Stanford Cardiovascular Institute, Stanford University, Stanford, CA, USA. ²¹Stanford Diabetes Research Center, Stanford University, Stanford, CA, USA. ²²Department of Medical Sciences, Molecular Epidemiology and Science for Life Laboratory, Uppsala University, Uppsala, Sweden. ²³Wellcome Centre for Human Genetics, University of Oxford, Oxford, UK. ²⁴Department of Biostatistics, University of Liverpool, Liverpool, UK. ²⁵Division of Musculoskeletal and Dermatological Sciences, University of Manchester, Manchester, UK. ²⁶Institute for Biomedicine,

Eurac Research, Affiliated Institute of the University of Lübeck, Bolzano, Italy. [27]Medical Sciences, Uppsala University Hospital, Uppsala, Sweden. [28]Nuffield Department of Medicine, Li Ka Shing Centre for Health Information and Discovery, Big Data Institute, University of Oxford, Oxford, UK. [29]Nuffield Department of Medicine, The Wellcome Centre for Human Genetics, University of Oxford, Oxford, UK. [30]The Broad Institute of MIT and Harvard, Cambridge, MA, USA. [31]Department of Medical Sciences, Uppsala University, Uppsala, Sweden. [32]Department of Cardiovascular Sciences, Cardiovascular Research Centre, Glenfield Hospital, University of Leicester, Leicester, UK. [33]NIHR Leicester Biomedical Research Centre, Glenfield Hospital, Groby Road, Leicester, UK. [34]Cardiovascular Department, Azienda Ospedaliera Universitaria Integrata of Trieste, Trieste, Italy. [35]Department of Clinical Physiology, Tampere University Hospital, Tampere, Finland. [36]Department of Clinical Physiology, Faculty of Medicine and Health Technology, Finnish Cardiovascular Research Center Tampere University, Tampere, Finland. [37]Department of Clinical Chemistry, Fimlab Laboratories, Tampere, Finland. [38]Department of Clinical Chemistry, Faculty of Medicine and Health Technology, Finnish Cardiovascular Research Center, Tampere University, Tampere, Finland. [39]Department of Cardiology, Heart Center, Tampere University Hospital, Tampere, Finland. [40]Department of Cardiology, Finnish Cardiovascular Research Center, Faculty of Medicine and Health Technology, Tampere University, Tampere, Finland. [41]Institute of Cardiovascular and Medical Sciences, College of Medical, Veterinary and Life Sciences, University of Glasgow, Glasgow, UK. [42]Institute of Cardiovascular and Medical Sciences, University of Glasgow, Glasgow, UK. [43]Division of Endocrinology, Diabetes, and Nutrition, University of Maryland School of Medicine, Baltimore, MD, USA. [44]Program for Personalized and Genomic Medicine, University of Maryland School of Medicine, Baltimore, MD, USA. [45]Cardiology Division, University Hospital, Basel, Switzerland. [46]Population Health Research Institute, McMaster University, Hamilton, ON, Canada. [47]Institute of Clinical Chemistry, Inselspital Bern, University Hospital, University of Bern, Bern, Switzerland. [48]Labormedizinisches Zentrum Dr. Risch, Vaduz, Liechtenstein. [49]Private University of the Principality of Liechtenstein, Triesen, Liechtenstein. [50]Department of Molecular Biology, Medical Biochemistry and Pathology, Laval University, Quebec, QC, Canada. [51]Department of Pediatrics, Tampere University Hospital, Tampere, Finland. [52]Department of Pediatrics, Faculty of Medicine and Health Technology, Tampere University, Tampere, Finland. [53]Department of Clinical Physiology and Nuclear Medicine, Turku University Hospital, Turku, Finland. [54]Research Centre of Applied and Preventive Cardiovascular Medicine, University of Turku, Turku, Finland. [55]Centre for Population Health Research, University of Turku and Turku University Hospital, Turku, Finland. [56]CARIM School for Cardiovascular Diseases, Maastricht Center for Systems Biology (MaCSBio), Department of Biochemistry, and Department of Physiology, Maastricht University, Maastricht, The Netherlands. [57]Department of Medical Informatics Erasmus MC, University Medical Center Rotterdam, Rotterdam, The Netherlands. [58]Genetic Epidemiology Unit, Department of Epidemiology, Erasmus University Medical Center, Rotterdam, The Netherlands. [59]Icelandic Heart Association, Kopavogur, Iceland. [60]Faculty of Medicine, University of Iceland, Reykjavik, Iceland. [61]Laboratory of Epidemiology and Population Sciences, National Institute on Aging, NIH, Baltimore, MD, USA. [62]School of Public Health, Department of Biostatistics, University of Michigan, Ann Arbor, MI, USA. [63]The Charles Bronfman Institute for Personalized Medicine, Icahn School of Medicine at Mount Sinai, New York, NY, USA. [64]The Mindich Child Health and Development Institute, Icahn School of Medicine at Mount Sinai, New York, NY, USA. [65]Jackson Heart Study, Department of Medicine, University of Mississippi Medical Center, Jackson, MS, USA. [66]Department of Data Science, University of Mississippi Medical Center, Jackson, MS, USA. [67]Department of Physiology and Biophysics, University of Mississippi Medical Center, Jackson, MS, USA. [68]DZHK (German Centre for Cardiovascular Research), Munich Heart Alliance, Munich, Germany. [69]Institute of Human Genetics, Helmholtz Zentrum München - German Research Center for Environmental Health, Neuherberg, Germany. [70]Institute of Human Genetics, Klinikum rechts der Isar, Technische Universität München, Munich, Germany. [71]Institute of Genetic Epidemiology, Helmholtz Zentrum München - German Research Center for Environmental Health, Neuherberg, Germany. [72]IBE, Faculty of Medicine, LMU Munich, Munich, Germany. [73]Department of Internal Medicine I (Cardiology), Hospital of the Ludwig-Maximilians-University (LMU) Munich, Munich, Germany. [74]German Center for Diabetes Research, Neuherberg, Germany. [75]Institute of Epidemiology, Helmholtz Zentrum München—German Research Center for Environmental Health, Neuherberg, Germany. [76]Research Unit of Molecular Epidemiology, Helmholtz Zentrum München—German Research Center for Environmental Health, Neuherberg, Germany. [77]Department of Twin Research and Genetic Epidemiology, Kings College London, London, UK. [78]NIHR Biomedical Research Centre at Guy's and St Thomas' Foundation Trust, London, UK. [79]Genomics plc, Oxford, UK. [80]Intermountain Heart Institute, Intermountain Medical Center, Murray, UT, USA. [81]Molecular Cardiology and Biophysics Division, Victor Chang Cardiac Research Institute, Darlinghurst, NSW, Australia. [82]Cardiology Department, St. Vincent's Hospital, Darlinghurst, NSW, Australia. [83]St Vincent's Clinical School, Faculty of Medicine, UNSW Sydney, Kensington, NSW, Australia. [84]Department of Cardiovascular Medicine, University of Iowa, Iowa City, IA, USA. [85]Laboratory for Molecular Cardiology, Department of Cardiology, The Heart Centre, Rigshospitalet, University Hospital of Copenhagen, Copenhagen, Denmark. [86]Department of Biomedical Sciences, University of Copenhagen, Copenhagen, Denmark. [87]Departments of Medicine, Pharmacology, and Biomedical Informatics, Vanderbilt University Medical Center, Nashville, TN, USA. [88]Department of Medicine, Vanderbilt University Medical Center, Nashville, TN, USA. [89]Department of Cardiology, Clinical Sciences, Wallenberg Center for Molecular Medicine, Lund University Diabetes Center, Lund University and Skane University Hospital, Lund, Sweden. [90]Cardiovascular Health Research Unit, Department of Medicine, University of Washington, Seattle, WA, USA. [91]Department of Biostatistics, University of Washington, Seattle, WA, USA. [92]Department of Epidemiology, University of Washington, Seattle, WA, USA. [93]Kaiser Permanente Washington Health Research Institute, Seattle, WA, USA. [94]Cardiology Division, University of Washington, Seattle, WA, USA. [95]Robertson Center for Biostatistics, University of Glasgow, Glasgow, UK. [96]Department of Cardiology, Leiden University Medical Center, Leiden, The Netherlands. [97]Einthoven Laboratory for Experimental Vascular Medicine, Leiden University Medical Center, Leiden, The Netherlands. [98]Institute of Health and Wellbeing, College of Medical, Veterinary and Life Sciences, University of Glasgow, Glasgow, UK. [99]Department of Internal Medicine, section of Gerontology and Geriatrics, Leiden University Medical Center, Leiden, The Netherlands. [100]DZHK (German Centre for Cardiovascular Research), Greifswald, Germany. [101]Department of Internal Medicine B - Cardiology, Pneumology, Infectious Diseases, Intensive Care Medicine, University Medicine Greifswald, Greifswald, Germany. [102]Interfaculty Institute for Genetics and Functional Genomics; Department of Functional Genomics; University Medicine and University of Greifswald, Greifswald, Germany. [103]Finnish Institute for Health and Welfare, Helsinki, Finland. [104]Institute for Molecular Medicine Finland (FIMM), HiLIFE, University of Helsinki, Helsinki, Finland. [105]Institute for Translational Genomics and Population Sciences and Department of Pediatrics, Los Angeles Biomedical Research Institute at Harbor-UCLA Medical Center, Torrance, CA, USA. [106]Cardiovascular Health Research Unit and Department of Epidemiology, University of Washington, Seattle, WA, USA. [107]Institute for Translational Genomics and Population Sciences, Los Angeles Biomedical Research Institute at Harbor-UCLA Medical Center, Torrance, CA, USA. [108]Department of Clinical Epidemiology, Leiden University Medical Center, Leiden, The Netherlands. [109]Department of Public Health and Primary Care, Leiden University Medical Center, Leiden, The Netherlands. [110]Department of Biomedical Sciences, University of Sassari, Sassari, Italy. [111]Laboratory of Genetics and Genomics, NIA/NIH, Baltimore, MD, USA. [112]Laboratory of Cardiovascular Science, NIA/NIH, Baltimore, MD, USA. [113]Population Sciences Branch, Division of Intramural Research, National Heart, Lung, and Blood Institute, Bethesda, MD, USA. [114]National Heart Lung and Blood Institute's and Boston University's Framingham Heart Study, Framingham, MA, USA. [115]Section of Computational Biomedicine, Department of Medicine, Boston University School of Medicine, Boston, MA, USA. [116]Center for Human Genetic Research and Cardiovascular Research Center, Harvard Medical School and Massachusetts General Hospital, Boston, MA, USA. [117]Department of Biostatistics, Boston University School of Public Health, Boston, MA, USA. [118]The Institute of Medical

Sciences, Aberdeen Biomedical Imaging Centre, University of Aberdeen, Aberdeen, UK. [119]Centre for Genomic and Experimental Medicine, Institute of Genetics & Molecular Medicine, University of Edinburgh, Western General Hospital, Edinburgh, UK. [120]Centre for Cognitive Ageing and Cognitive Epidemiology, University of Edinburgh, Edinburgh, UK. [121]Division of Population Health and Genomics, Ninewells Hospital and Medical School, University of Dundee, Dundee, UK. [122]Department of Epidemiology Erasmus MC, University Medical Center Rotterdam, Rotterdam, The Netherlands. [123]Human Genotyping Facility Erasmus MC University Medical Center Rotterdam, Rotterdam, The Netherlands. [124]Fred Hutchinson Cancer Research Center, Division of Public Health Sciences, Seattle, WA, USA. [125]Division of Endocrinology, Diabetes and Metabolism, Ohio State University, Columbus, OH, USA. [126]Department of Epidemiology, University of Washington, Seattle, WA, USA. [127]Departments of Epidemiology and Medicine, Gillings School of Global Public Health and School of Medicine, University of North Carolina, Chapel Hill, NC, USA. [128]Department of Epidemiology, Rollins School of Public Health, Emory University, Atlanta, GA, USA. [129]McKusick-Nathans Institute of Genetic Medicine, Johns Hopkins University School of Medicine, Baltimore, MD, USA. [130]Human Genetics Center, University of Texas Health Science Center at Houston, Houston, TX, USA. [131]Cardiology, Geneva University Hospitals, Geneva, Switzerland. [132]Epidemiological Cardiology Research Center, Wake Forest School of Medicine, Winston-Salem, NC, USA. [133]Department of Epidemiology, University of North Carolina, Chapel Hill, NC, USA. [134]Carolina Population Center, University of North Carolina, Chapel Hill, NC, USA. [135]Division of Pharmaceutical Outcomes and Policy, University of North Carolina, Chapel Hill, NC, USA. [136]Barts Heart Centre, St Bartholomews Hospital, London, UK. [137]Institute of Cardiovascular Science, University College London, London, UK. [138]Department of Mechanical Engineering, University College London, London, UK. [139]Department of Medicine, Landspitali University Hospital, Reykjavik, Iceland. [140]School of Engineering and Natural Sciences, University of Iceland, Reykjavik, Iceland. [141]Section of Cardiovascular Medicine and Section of Preventive Medicine, Department of Medicine, Boston University School of Medicine, Boston, MA, USA. [142]Department of Epidemiology, Boston University School of Public Health, Boston, MA, USA. [143]Cardiac Arrhythmia Service, Massachusetts General Hospital, Boston, MA, USA. [144]Genetics Research Centre, Molecular and Clinical Sciences Institute, St George's, University of London, London, UK. [145]These authors contributed equally: Ioanna Ntalla, Lu-Chen Weng. [146]These authors jointly supervised this work: Steven A. Lubitz, Patricia B. Munroe. ✉email: slubitz@mgh.harvard.edu; p.b.munroe@qmul.ac.uk

