## [Peer Review File · Nature Communications]

Reviewers' Comments:

Reviewer #1:

Remarks to the Author:

Ntalla et al. performed a GWAS meta-analysis of electrocardiographic PR intervals and identified a total of 210 significant loci, of which 149 are novel. The authors also carried out bioinformatics and in silico functional annotations, and polygenic risk score (PRS) analyses, to explore the functional relevance of the associated loci. The study is comprehensive and provides substantial insight into the polygenic basis of cardiac conduction. There are several points, which the authors should take into consideration.

Major points:

1. While the sample size is relatively large (N=293051 for multi-ancestry and N=271570 for European ancestry) as a whole, there seems to be insufficiency in the validation/replication of association signals. The distinction between multi-ancestry meta-analysis and ancestry-specific meta-analysis should be made clearer. Eight of 149 novel loci failed to satisfy a genome-wide significance level ($P < 5E-8$) in the multi-ancestry meta-analysis. If a priori study design is a multi-ancestry meta-analysis, such association signals detectable only in the European ancestry analysis should be regarded as post hoc and validated independently. Also, since there is no validation/replication stage in the present study, it is appropriate to demonstrate reproducibility of some association signals, especially for those showing not highly significant (e.g., $1E-8 < P < 5E-8$) association, considering that low-frequency and rare variants are included in the meta-analysis.
2. In the similar vein, more attention had better be paid for the distinction between multi-ancestry analysis and ancestry-specific analysis of eQTL and PRS. In particular, it has been reported that the generalizability of PRS for multifactorial diseases and quantitative traits is limited across diverse populations (PMID: 30926966). Since European-descent individuals from UK Biobank are used to calculate GRS, it is more straightforward to focus on European-specific analysis of PRS.
3. Bioinformatics and in silico functional annotations help to provide a list of candidate causative genes in the associated loci but the eventual nomination of each candidate should be made carefully. There seem to be a bit over-emphasis on functional relevance of individual candidate genes throughout the main text.
4. In the part of PRS analysis, the authors claimed to report nominal associations without establishing a pre-specified significance threshold (lines 630-631). Although nominally significant ($P < 0.05$), the association between PRS for PR interval prolongation and a reduced risk of non-ischemic cardiomyopathy and coronary heart disease had better be interpreted carefully (lines 387-389), in particular, without biological plausibility. Some degree of adjustment for multiple testing should be made, even though there are substantial correlations between tested traits known to be associated with cardiac conduction.
5. For the association between PRS for PR interval prolongation and AF risk, the reviewer recommends the authors to perform more detailed analysis for the purpose of identifying distinct pathophysiologic mechanisms. Apart from PRS analysis, is it possible to evaluate effect sizes of PR-interval associated variants on AF risk?

Minor points:

6. Figure 1 should be revised to show two types of discovery stage analysis, i.e., multi-ancestry meta-analysis and ancestry-specific meta-analysis.
7. For the 64 known loci, please indicate the degree of sample overlap between the present study and previous GWAS in Suppl. Table 7.
8. In Figure 4, please indicate the thresholds for dotted lines.

Reviewer #2:

Remarks to the Author:

Dr Ntalla and colleagues have reported here a large GWAS of PR interval from ECG data. They

followed this with bioinformatics investigation of loci using many available tools and databases. The final aspect was performing a phewas of the PR interval PRS in individuals not included in the original discovery analysis.

I think this is an interesting and rigorous study of the genetic architecture of this trait in a large meta-analysis dataset. They did many of the current bioinformatic follow-up work to provide insights into the underlying biology of this trait including eQTL analysis in GTEx, colocalization with non-coding elements, enrichment in DEPICT and pathway analysis in IPA, then a phewas to look for associated clinical outcomes. Their methods are statistically rigorous and adhere to the standards in the field.

The only suggestion I would make is that it would have been helpful to see the authors run the eQTL and PredicXscan analyses using either a negative-control tissue or in a variety of other tissue-types, in addition to their tissues of interest. It is hard to interpret these results when it is not clear if the associations they are seeing are tissue-specific or not.

I would recommend this manuscript for publication if the authors address this point.

Reviewer #3:

Remarks to the Author:

Dr. Ntallia and colleagues have submitted a manuscript that studies the relationship between genetic variants at a genome-wide scale and the PR interval in a combination of several cohorts and multiple ancestries. The investigators performed a meta analysis of summary data from individual cohort analyses from an additive model adjusted for several demographic and biometric factors as well as principle components to adjust for population stratification. The authors found that there were 149 novel loci associated with the PR interval and that these, in combination with established variants, explain 62.6% of the heritability of PR interval. They further describe the pathophysiologic processes enriched in these PR-related genes.

The PR interval is a fundamental biologic cardiac property that has a relationship to several underlying physiologic functions and disease states. The genomic architecture of the PR interval is therefore of significant interest. The authors are to be commended for such a large international collaboration. The combination of multiple racial/ethnic groups adds to the novelty and appears to have contributed to the discovery many new loci.

I do have several comments and questions that I ask the authors will address and that I hope will strengthen this manuscript.

Major Comments:

Rather than use separate discovery and replication cohorts, the authors use a single discovery study design and only considered those with genomewide significance in 60% of the max sample size. The justification for this was based on such a large sample size. Can the authors explain how this is more robust compared to using only 60% of the sample size for discovery and 40% for replication? Are there references to this approach, in particular justifying how the selection of 60% was chosen? Was this 60% of the cohorts, or 60% of the participants? While this approach may be reasonable given such a large overall sample size, note that the non-European groups were relatively small, and also note that there may be a very small number of cohorts contributing most of the non-European participants. Was there equal distribution of non-European ethnic groups in the 60% sample compared to the entire cohort? How many variants reached genomewide significance in the 60% sample, but not in the full analysis?

The number of novel variants is quite large – larger than the number of previously reported

variants. Was this a function of just the absolute increase in number of participants included, or a function of combining the different racial/ethnic groups. To get at this question, can the authors tell us how many total novel variants would have been discovered had they only studied European ancestral groups? The authors note that there are 6 novel variants in the European only analyses, but this wasn't entirely clear. Does this mean that 6 novel variants were found in the European only analysis that were not found in the overall meta-analysis, but there were many other novel variants that were found in both the European only and in the combined ancestry analyses. If so, it would be helpful to know how many of the novel variants in all would have been found in the European only analysis, and importantly, how many novel variants were discovered because of the addition of the other racial/ethnic cohorts.

The assumption in combining the different racial/ethnic cohorts is that variants have a homogeneous effect in all racial ethnic groups and that by combining the ancestries, those with marginal significance now become significant. However, since the European group is significantly larger, variants with European-specific significance would remain significant even if the effects are smaller in the non-European cohorts. How many of the variants had heterogeneous effects across ancestries/ethnicities?

For adjustment of population stratification, were the PCs used cohort-specific or was this standardized across each racial/ethnic group. How many PCs were used in the models, and was this number uniform across all analyses or derived independently by each cohort?

The authors report at least one rare novel variant that was associated with PR interval. Note that the cutoff for genomewide significance is based on common variation ($MAF \sim >1\%$) across the genome in European cohorts. There is more common variation in non-European, particularly African, ancestral populations. There is even more variation once the $MAF < 1\%$. Can the authors justify their use of this cutoff for the discovery of the rare variant reported?

Similarly, I do have another concern regarding multiple hypothesis testing. Again, Bonferroni correction is used for multiple-hypothesis testing in one analysis using $\sim 1M$ representative common variants. However, the authors here performed several analyses (all cohorts, European only, African only, Hispanic only, Brazilian only, autosomal, non-autosomal). The opportunity for discovery here is additive, but the Bonferroni threshold for significance did not vary. Granted, most of the novel discovery was made in the overall group and the European only analysis, but by simply looking at different groups adds to the risk of false discovery. Do the authors have any justification for not adjusting further the multiple hypothesis threshold?

I'm not sure the polygenic risk score here adds significantly. Note that the PR interval itself is not a pathogenic entity, rather itself an endophenotype. While interesting that the PGRS is associated with a few cardiovascular diseases, what is the added value of the PGRS on top of the PR interval, which itself is easy to measure. Is there any added value? Also, can the authors explain why they believe the PGRS was inversely related to AF, while the PR is typically positively related to AF?

Minor Comments

In regards to the statement that there are several monogenic-linked genes that were found in/near the loci associated with PR prolongation, I would point out that many of these genes in actuality have weak evidence for their monogenic roles, except for HCN4 and RYR2 as the authors point out.

The authors point out that there were a number of novel missense variants, implying that these were the causative variants. While this is highly likely, I think it is still important to point out that these variants are still only a representation of multiple variants in high LD and that the causative variant(s) may not be those particular missense variants.

The authors explain there were several exclusion criteria, including use of certain medications. Was medication data reliably available in all cohorts? If not available, was the assumption that the participants were not on these medications. For exclusion criteria, did the authors mean those with paced rhythms were excluded, or those with pacemakers? Subtle but important distinction. The latter would be per participant history which may not be available from all cohorts.

The transcriptome analysis though interesting, does not add significantly to the primary findings. Most salient, the findings of the transcripts not present in the primary analysis are not replicated and their importance should be minimized.

Why did the authors put both height and BMI in the primary models? Along these lines, why not (height + weight) or (height + weight + BMI)?

Manuscript NCOMMS-19-26194-T³

Multi-ancestry GWAS of the electrocardiographic PR interval identifies 210 loci underlying cardiac conduction

Note to all reviewers:

As part of the revision of our manuscript we have made some edits to Supplementary Files which are not fully listed in our response to the Reviewers. These include additional information being added to legends for clarity, demarcation of novel findings and results from secondary analyses. In our revised files, we have indicated changes either as track changes in Word files or in red text in the Supplementary Tables.

We use yellow highlighting in our response to reviewers to indicate the changes made to the main manuscript.

Reviewers' comments:

Reviewer #1 (Remarks to the Author):

Major points:

1. While the sample size is relatively large (N=293051 for multi-ancestry and N=271570 for European ancestry) as a whole, there seems to be insufficiency in the validation/replication of association signals. The distinction between multi-ancestry meta-analysis and ancestry-specific meta-analysis should be made clearer. Eight of 149 novel loci failed to satisfy a genome-wide significance level ($P < 5E-8$) in the multi-ancestry meta-analysis. If a priori study design is a multi-ancestry meta-analysis, such association signals detectable only in the European ancestry analysis should be regarded as post hoc and validated independently. Also, since there is no validation/replication stage in the present study, it is appropriate to demonstrate reproducibility of some association signals, especially for those showing not highly significant (e.g., $1E-8 < P < 5E-8$) association, considering that low-frequency and rare variants are included in the meta-analysis.

Authors' reply:

We thank the reviewer for this comment. We now make a clear distinction between the multi-ancestry meta-analysis, which is our primary meta-analysis, and ancestry specific meta-analyses. We claim as novel only the loci that reached genome-wide significance in our multi-ancestry meta-analysis (141 loci). We also report results for the 8 loci that reached genome-wide significance in the European meta-analysis and were not previously reported, and indicate that these results are from secondary analysis. We took forward variants from all 149 loci for bioinformatics analysis and annotation, and the 8 loci from the European meta-analysis are now clearly indicated in tables and legends. We have accordingly revised the main text and supplementary material in the manuscript. Regarding our approach, given our access to a large sample size we decided at the outset to undertake a one-stage big discovery GWAS meta-analysis and use the genome-wide significance threshold ($P < 5 \times 10^{-8}$). We performed a strict quality control on GWAS results from each study and prefiltered variants of low quality. To ensure robustness of loci we then considered for discovery only variants that were present in at least 60% of our maximum sample size in the GWAS meta-analysis results ($N_{\text{samples}} \geq 175,831$). All of our novel loci discovered in the multi-ancestry meta-analysis were discovered in a sample greater than 212,667 individuals. The figure below shows the proportion of the total sample size analysed for all lead variants at the 141 novel loci. There are 20 loci across the allelic range with $1 \times 10^{-8} < P < 5 \times 10^{-8}$ and only 2 loci with variants with a MAF less than 5%.

There is precedence for similar one-stage large discovery studies of comparable sample sizes to ours reported by other studies in the literature. For example, recently Tin et al. [PMID: 31578528] in GWAS meta-analyses of serum urate levels in 457,690 individuals followed a similar approach to ours. For each meta-analysis result that they performed, they excluded variants that were present in $< 50\%$ of the studies and they also defined genome-wide significance at $P < 5 \times 10^{-8}$. They reported 183 genome-wide significant loci, of which 147 were previously unknown. Wyss et al [PMID: 30061609] in a meta-analysis of pulmonary function in 90,715 individuals primarily of European ancestry (67%), following meta-analysis they excluded variants with less than one-third the total sample size or less than the sample size of the largest study for a given meta-analysis and declared significance at $P < 5 \times 10^{-8}$.

Manuscript changes:

Title

Page 1:

“Multi-ancestry GWAS of the electrocardiographic PR interval identifies 202 loci underlying cardiac conduction”

Introductory paragraph

Page 14

The electrocardiographic PR interval reflects atrioventricular conduction, and is associated with conduction abnormalities, pacemaker implantation, atrial fibrillation (AF), and cardiovascular mortality^{1,2}. We performed multi-ancestry (N=293,051) genome-wide association (GWAS) meta-analysis for the PR interval, discovering 202 loci of which 141 are novel.

Main text

Page 15:

We identified a total of 202 genome-wide significant loci ($P < 5 \times 10^{-8}$) in a multi-ancestry analysis, of which 141 were not previously reported (Table 1, Fig. 2, Supplementary Fig. 1 and 2). We considered for discovery only variants present in >60% of the maximum sample size in the GWAS summary results, a filtering criterion used to ensure robustness of associated loci (median proportion of sample size included in analyses for lead variants 1.0, interquartile range 0.99-1.00; Online Methods). There was strong support for all 64 previously reported loci (61 at $P < 5 \times 10^{-8}$ and 3 at $P < 1.1 \times 10^{-4}$; Supplementary Tables 6-7). In a secondary analysis among the European ancestry subset, a total of 127 loci not previously reported reached genome-wide significance (Supplementary Tables 4-5, Supplementary Figs. 1-4), of which lead variants at 8 loci were borderline genome-wide significant ($P < 9.1 \times 10^{-7}$) in our multi-ancestry meta-analysis. No loci that were not previously reported were genome-wide significant in African or Hispanic/Latino ancestry meta-analyses (Supplementary Table 8, Supplementary Fig. 1 and 3) or X chromosome meta-analyses (Supplementary Fig. 5). In sensitivity analyses, we examined the rank-based inverse normal transformed residuals of PR interval. Results of absolute and transformed trait

meta-analyses were highly correlated ($\rho > 0.94$, **Supplementary Tables 5, 9-10, Supplementary Fig. 6-7**).

Page 19:

To summarise, in meta-analyses of nearly 300,000 individuals we identified 202 loci, of which 141 were novel, underlying cardiac conduction as manifested by the electrocardiographic PR interval.

Online Methods

Page 25:

We aggregated summary level associations between genotypes and absolute PR interval from all individuals (N=293,051), and only from Europeans (N=271,570), African Americans (N=8,173), and Hispanic/Latinos (N=12,823) using a fixed-effects meta-analysis approach implemented in METAL (release on 2011/03/25)⁴⁴. We considered as primary our multi-ancestry meta-analysis, and ancestry specific meta-analyses as secondary.

Pages 25-26:

Given the large sample size we undertook a one-stage discovery study design. To ensure the robustness of this approach we considered for discovery only variants reaching genome-wide significance ($P < 5 \times 10^{-8}$) present in at least 60% of the maximum sample size (N_{\max}) in our GWAS summary results. We declared as novel any variants mapping outside the 64 loci previously reported (**Supplementary Note, Supplementary Table 6**) in our primary meta-analysis (multi-ancestry meta-analysis). We are also reporting genome-wide significant variants not previously reported in ancestry specific meta-analysis (secondary meta-analyses). We grouped genome-wide significant variants into independent loci based on both distance ($\pm 500\text{kb}$) and linkage disequilibrium (LD, $r^2 < 0.1$) (**Supplementary Note**). We assessed heterogeneity in allelic effect sizes among studies contributing to the meta-analysis and among ancestral groups by the I^2 inconsistency index⁴⁵ for the lead variant in each novel locus. LocusZoom⁴⁶ was used to create region plots of identified loci. We are declaring as novel discoveries any genome-wide significant loci from our primary meta-analysis. However we considered ancestry-specific loci for annotation and downstream analyses. The results from secondary analyses are specifically indicated in the Supplementary Tables.

Meta-analyses (multi-ancestry [N=282,128], European only [N=271,570], and African [N=8,173]) of rank-based inverse normal transformed residuals of PR interval were also performed (sensitivity meta-analyses). Because not all studies contributed summary level association statistics of the transformed PR interval, we considered as primary the multi-ancestry meta-analysis of absolute PR interval for which we achieved the maximum sample size. Loci that met our significance criteria in the meta-analyses of transformed PR interval were not taken forward for downstream analyses.

Table 1

Pages 31-35:

The results from the European meta-analysis were removed from Table 1, only novel loci are presented.

2. In the similar vein, more attention had better be paid for the distinction between multi-ancestry analysis and ancestry-specific analysis of eQTL and PRS. In particular, it has been reported that the generalizability of PRS for multifactorial diseases and quantitative traits is limited across diverse populations (PMID: 30926966). Since European-descent individuals from UK Biobank are used to calculate GRS, it is more straightforward to focus on European-specific analysis of PRS.

Authors' reply:

We thank the reviewer for their suggestion. We have updated our results and method sections and now report the European PRS results in the main text.

Manuscript changes:

Main text

Pages 18-19:

Finally, we evaluated associations between genetic predisposition to PR interval duration and 16 cardiac phenotypes chosen *a priori* using ~309,000 unrelated UKB European participants not included in our meta-analyses²¹. We created a polygenic risk score (PRS) for PR interval using the **European ancestry** meta-analysis results (**Fig. 4, Supplementary Table 22**). Genetically determined PR interval prolongation was associated with higher risk of distal conduction disease (atrioventricular block; odds ratio [OR] per standard deviation 1.11, $P=7.02 \times 10^{-8}$) and pacemaker implantation (OR 1.06, $P=1.5 \times 10^{-4}$). In contrast, genetically determined PR interval prolongation was associated with reduced risk of AF (OR 0.95, $P=4.30 \times 10^{-8}$) and was marginally associated with a reduced risk of atrioventricular pre-excitation (Wolff-Parkinson-White syndrome; OR 0.85, $P=0.0032$). Results were similar when using a PRS derived using the **multi-ancestry** ancestry meta-analysis results (**Supplementary Fig. 11, Supplementary Table 22**).

Online Methods

Pages 30-31:

Associations between genetically determined PR interval and cardiovascular conditions

We examined associations between genetic determinants of atrioventricular conduction and candidate cardiovascular diseases in unrelated individuals of European ancestry from UK Biobank (N~309,000 not included in our GWAS meta-analyses) by creating PRSs for PR interval based on our GWAS results. We derived two PRSs. One was derived from the **European ancestry** meta-analysis results, and the other from the **multi-ancestry** meta-analysis results. We used the LD-clumping feature in PLINK v1.90⁶¹ ($r^2=0.1$, window size=250kb, $P=5 \times 10^{-8}$) to select variants for each PRS. Referent LD structure was based on 1000G **European only, and** all ancestry data. In total, we selected **582 and** 743 variants from European only **and multi-ancestry** meta-analysis results, respectively. We calculated the PRSs for PR interval by summing the dosage of PR interval prolonging alleles weighted by the corresponding effect size from the meta-analysis results. A total of **581** variants for the PRS derived from **European** results and **743** variants for the PRS derived from **multi-ancestry** results (among the variants with imputation quality >0.6) were included in our PRS calculations.

3. Bioinformatics and in silico functional annotations help to provide a list of candidate causative genes in the associated loci but the eventual nomination of each candidate should be made carefully. There seem to be a bit over-emphasis on functional relevance of individual candidate genes throughout the main text.

Authors' reply:

We have reviewed and edited the text in order to be clear that the candidate genes we discuss are “candidates” at the locus.

Manuscript changes:

Main text:

Page 18:

Bioinformatics and *in silico* functional annotations for potential candidate genes at the 149 loci are summarised in **Supplementary Tables 18-19**.

Page 20:

Common variants in/near genes associated with monogenic arrhythmia syndromes were also observed, suggesting these genes may also affect atrioventricular conduction and cardiovascular pathology in the general population. Apart from *DSP*, *DES* and *GJA5* discussed above, our analyses indicate 2 additional candidate genes (*HCN4* and *RYR2*). *HCN4* encodes a component of the hyperpolarization-activated cyclic nucleotide-gated potassium channel which specifies the sinoatrial pacemaker “funny” current, and is implicated in sinus node dysfunction, AF, and left ventricular noncompaction³⁴⁻³⁶. *RYR2* encodes a calcium channel component in the cardiac sarcoplasmic reticulum and is implicated in catecholaminergic polymorphic ventricular tachycardia³⁷.

4. In the part of PRS analysis, the authors claimed to report nominal associations without establishing a pre-specified significance threshold (lines 630-631). Although nominally significant ($P < 0.05$), the association between PRS for PR interval prolongation and a reduced risk of non-ischemic cardiomyopathy and coronary heart disease had better be interpreted carefully (lines 387-389), in particular, without biological plausibility. Some degree of adjustment for multiple testing should be made, even though there are substantial correlations between tested traits known to be associated with cardiac conduction.

Authors' reply:

We agree we should be cautious in interpreting nominal associations without adjustment for multiple testing. Accordingly, we have edited the text and now only report associations that pass the significance threshold after Bonferroni correction.

Manuscript changes:

Main text:

Page 14:

We showed that polygenic predisposition to PR interval duration is an endophenotype for cardiovascular disease risk, including distal conduction disease, AF, and atrioventricular pre-excitation.

Pages 18-19:

Finally, we evaluated associations between genetic predisposition to PR interval duration and 16 cardiac phenotypes chosen *a priori* using ~309,000 unrelated UKB European participants not included in our meta-analyses²¹. We created a polygenic risk score (PRS) for PR interval using the European ancestry meta-analysis results (**Fig. 4, Supplementary Table 22**). Genetically determined PR interval prolongation was associated with higher risk of distal conduction disease (atrioventricular block; odds ratio [OR] per standard deviation 1.11, $P = 7.02 \times 10^{-8}$) and pacemaker implantation (OR 1.06, $P = 1.0 \times 10^{-4}$). In contrast, genetically determined PR interval prolongation was associated with reduced risk of AF (OR 0.95, $P = 4.30 \times 10^{-8}$) and marginally associated with a reduced risk of atrioventricular pre-excitation (Wolff-Parkinson-White syndrome; OR 0.85, $P = 0.0032$). Results were similar when using a

PRS derived using the **multi-ancestry** ancestry meta-analysis results (**Supplementary Fig. 11, Supplementary Table 22**).

Page 31:

Given correlation between traits, we **set** significance threshold at $P < 3.13 \times 10^{-3}$ after **Bonferroni correction** ($P < 0.05/16$) for the **number of analyses performed** and **also** report nominal associations ($P < 0.05$).

Pages 42-43:

Figure 4 Bubble plot of phenome-wide association analysis of **European** ancestry PR interval polygenic risk score.

The **polygenic** risk score was derived from the **European ancestry** meta-analysis. Orange circles indicate that polygenic **predisposition to longer** PR interval is associated with an increased risk of the condition, whereas blue circles indicate that **polygenic predisposition to longer PR interval** is associated with lower risk **of the condition**. The darkness of the colour reflects the effect size (odds ratio, OR) per 1 standard deviation increment of the polygenic risk score. Given correlation between traits, **we set** significance threshold at $P < 3.13 \times 10^{-3}$ after Bonferroni correction ($P < 0.05/16$; dotted line) for the number of analyses performed and also report nominal associations ($P < 0.05$; dashed line).

Supplementary note:

Pages 46-47:

Supplementary Figure 12 Bubble plot of phenome-wide association analysis of **multi-ancestry** PR interval polygenic risk score.

The polygenic risk score was derived from the multi-ancestry meta-analysis. Orange circles indicate that polygenic predisposition to longer PR interval is associated with an increased risk of the condition, whereas blue circles indicate that polygenic predisposition to longer PR interval is associated with lower risk of the condition. The darkness of the color reflects the effect size (odds ratio, OR) per 1 standard deviation increment of the polygenic risk score. Given correlation between traits, we set significance threshold at $P < 3.13 \times 10^{-3}$ after Bonferroni correction ($P < 0.05/16$; dashed line) for the number of analyses performed and also report nominal associations ($P < 0.05$; dotted line).

5. For the association between PRS for PR interval prolongation and AF risk, the reviewer recommends the authors to perform more detailed analysis for the purpose of identifying distinct pathophysiologic mechanisms. Apart from PRS analysis, is it possible to evaluate effect sizes of PR-interval associated variants on AF risk?

Authors' reply:

Yes, this is possible and following your recommendation we have performed a look-up of all PR interval related variants for AF risk from previously reported AF papers [PMIDs: 30061737, 29892015]. We observe discordant directions of individual genetic variants for PR and AF risk which imply potentially distinct pathophysiologic mechanisms of cardiac arrhythmia. We have included the summarized look-up plots as a new supplementary figure.

Manuscript changes:

Main text:

Page 18:

Using a prior GWAS of AF for reference^{20,21}, we identified variants with shared associations between PR interval duration and AF risk (**Supplementary Fig. 11**).

Page 29:

For AF, we summarized the results of lead PR interval variants for PR interval and their associations with AF risk from two recently published GWASs.^{20,21} We included variants in high linkage disequilibrium with lead PR variants ($r^2 > 0.7$).

Supplementary note:

Pages 44-45:

Supplementary Figure 11 Associations between lead PR interval variants (205 single nucleotide polymorphisms) with atrial fibrillation (AF) risk from recently published AF GWASs.

The X axis refers to $-\log_{10}$ P-value for PR interval. Y axis refers to $-\log_{10}$ P-value for AF risk. Red/orange color indicates the same direction of effect for PR interval and AF risk, while blue colour indicates the opposite direction of effect for PR interval and AF risk. The different color scheme shows different odds ratio of AF risk. The nearest gene names are labelled if variants were genome-wide significantly associated with both traits. Upper panel (a) is the look-up results from Roselli et al.², and lower panel (b) is the look-up results from Nielsen et al.²⁹

Minor points:

6. Figure 1 should be revised to show two types of discovery stage analysis, i.e., multi-ancestry meta-analysis and ancestry-specific meta-analysis.

Authors' reply:

As discussed above, the revised submission makes a clear distinction between the multi-ancestry meta-analysis, which is our primary meta-analysis, and ancestry specific meta-analyses. We have updated main Figure 1, and revised the text accordingly.

7. For the 64 known loci, please indicate the degree of sample overlap between the present study and previous GWAS in Suppl. Table 7.

Authors' reply:

Samples from 23 of the 40 studies that contributed to the most recent published GWAS for PR interval were included in our meta-analysis (PMID: 30046033), thus there is some overlap. Of these one study (DeCODE) contributed 9,000 samples to the prior study, while over 80,000 samples from this study were included in our study. We have indicated in a revised Supplementary Table 1 the studies that were included in the previous GWAS for PR interval.

8. In Figure 4, please indicate the thresholds for dotted lines.

Authors' reply:

We have added footnotes for the thresholds in both Figure 4 and Supplementary Figure 12 and updated figure legends accordingly.

Manuscript changes:

Main text:

Page 42:

Figure 4 Bubble plot of phenome-wide association analysis of **European** ancestry PR interval polygenic risk score.

The **polygenic** risk score was derived from the **European ancestry** meta-analysis. Orange circles indicate that polygenic **predisposition to longer PR intervals** is associated with an increased risk of the condition, whereas blue circles indicate that **polygenic predisposition to longer PR intervals** is associated with lower risk **of the condition**. The darkness of the colour reflects the effect size (odds ratio, OR) per 1 standard deviation increment of the polygenic risk score. Given correlation between traits, **we set significance threshold at $P < 3.13 \times 10^{-3}$ after Bonferroni correction ($P < 0.05/16$; dotted line)** for the number of analyses performed and also report **nominal associations ($P < 0.05$; dashed line)**.

Supplementary note

Pages 46:

Supplementary Figure 12 Bubble plot of phenome-wide association analysis of **multi-ancestry** PR interval polygenic risk score.

The **polygenic** risk score **was** derived from the **multi-ancestry** meta-analysis. Orange circles indicate that polygenic **predisposition to longer PR intervals** is associated with an increased risk of the condition, whereas blue circles indicate that polygenic **predisposition to longer PR intervals** is associated with lower risk of the condition. The darkness of the color reflects the **effect size (odds ratio, OR)** per 1 **standard deviation** increment of the polygenic risk score. Given correlation between traits, **we set significance threshold at $P < 3.13 \times 10^{-3}$ after Bonferroni correction ($P < 0.05/16$; dashed line)** for the number of analyses performed and **also** report nominal associations ($P < 0.05$; **dotted line**).

Reviewer #2 (Remarks to the Author):

Dr Ntalla and colleagues have reported here a large GWAS of PR interval from ECG data. They followed this with bioinformatics investigation of loci using many available tools and databases. The final aspect was performing a phewas of the PR interval PRS in individuals not included in the original discovery analysis.

I think this is an interesting and rigorous study of the genetic architecture of this trait in a large meta-analysis dataset. They did many of the current bioinformatic follow-up work to provide insights into the underlying biology of this trait including eQTL analysis in GTEx, colocalization with non-coding elements, enrichment in DEPICT and pathway analysis in IPA, then a phewas to look for associated clinical outcomes. Their methods are statistically rigorous and adhere to the standards in the field.

The only suggestion I would make is that it would have been helpful to see the authors run the eQTL and PrediXscan analyses using either a negative-control tissue or in a variety of other tissue-types, in addition to their tissues of interest. It is hard to interpret these results when it is not clear if the associations they are seeing are tissue-specific or not.

I would recommend this manuscript for publication if the authors address this point.

Authors' reply:

We have now performed eQTL and S-PrediXscan analyses including spleen as a negative-control tissue as the reviewer suggested. We also took the opportunity during the revision of our manuscript to perform colocalization analysis using COLOC (PMIDs: 19039033, 24227294, 24830394) to determine if PR interval associated variants were colocalised with eQTLs in left ventricle (LV), right atrial appendage (RAA) and spleen. We observed colocalization and tissue specificity for many PR interval variants at both novel and previously reported loci in LV and RAA tissue only, and we now include these results in the paper.

Manuscript changes:

Main text:

Page 17:

PR interval lead variants (or best proxy [$r^2 > 0.8$]) at **43** novel and 23 previously reported loci were significant cis-eQTLs (at a 5% false discovery rate (FDR) in left ventricle (LV) and right atrial appendage (RAA) tissue samples from the Genotype-Tissue Expression (GTEx) project¹⁸. Variants at **13** and **6** previously reported loci were eQTLs in spleen, which was used as negative control tissue (**Supplementary Table 13**). The PR interval associations and eQTLs colocalised at **31** novel loci and **14** previously reported loci (posterior probability [PP] > 75%). Variants at **9** novel loci were significant eQTLs only in LV and RAA tissues with consistent directionality of gene expression.

In an exploratory analysis, we also performed a transcriptome-wide analysis to evaluate associations between PR interval duration and predicted gene expression in LV and RAA. We identified **113** genes meeting our significance threshold ($P < 3.1 \times 10^{-6}$, after Bonferroni correction), of which **91** were localised at PR interval loci (within 500kb from a lead variant; **Supplementary Table 14, Supplementary Fig. 8**). Longer PR interval duration was associated with decreased levels of

predicted gene expression for 57 genes, and increased levels for 56 genes (Fig. 3). In spleen tissues, only 31 gene expression-PR interval associations were detected, and 19 of them did not overlap with the findings in heart tissues.

Page 28:

For both eQTL and S-PrediXcan assessments, we additionally included spleen tissue (N=146) as a negative control. In total, we tested 5,366, 5,977, and 4,598 associations in LV, RAA, and spleen, respectively. Significance threshold of S-PrediXcan was set at $P = 3.1 \times 10^{-6}$ ($=0.05/(5,977+5,366+4,598)$) to account for multiple testing. In order to determine whether the GWAS identified loci were colocalized with the eQTL analysis, we performed genetic colocalization analysis for eQTL and S-PrediXcan identified gene regions, using the Bayesian approach in COLOC package (R version 3.5). Variants located within the same identified gene regions were included. We set the significant threshold for the PP (two significant associations sharing a common causal variant) at >75%.

Page 41:

Figure 3 Plausible candidate genes of PR interval from S-PrediXcan.

Diagram of standard electrocardiographic intervals and the heart. The electrocardiographic features are illustratively aligned with the corresponding cardiac conduction system structures (orange) reflected on the tracing. The PR interval (labeled) indicates conduction through the atria, atrioventricular node, His bundle, and Purkinje fibers. Right: The tables show 113 genes whose expression in the left ventricle (N=272) or right atrial appendage (N=264) was associated with PR interval duration in a transcriptome-wide analysis using S-PrediXcan and GTEX v7. Displayed genes include those with significant associations after Bonferroni correction for all tested genes ($P < 3.1 \times 10^{-6}$). Longer PR intervals were associated with increased predicted expression of 56 genes (blue) and reduced expression of 57 genes (orange).

Gene expression related to longer PR

ACP6	DNM1P51	MYO15A	TCTN3
AL590822.1	EDN2	NPIPA5	TMEM182
ALPK3	EFNA1	NUDT13	TPMT
ATP5D	FADS1	PDZRN3	TRAK1
BMPR1A	FAM211B	PHACTR1	TRIP4
C11orf1	FAT1	RP11-29H23.5	TTC18
CALHM2	FKBP7	RP11-399K21.1	VDAC2
CAMK2D	FUT11	RP11-3B7.1	VPREB3
CCDC36	GBAP1	RP4-764O22.2	XIRP1
CDH13	HMGAI1P5	RPSA	ZCCHC24
CEFIP	IFRD2	SLC25A26	ZNF503-AS1
CFDP1	KCND3	SLC6A6	
CHRM2	KDM1B	SLK	
DAG1	LRCH1	SNX1	
DEK	MSTO2P	SYNPO2L	

Gene expression related to shorter PR

AC011747.4	IL17D	PLCD1	SPATA20
AC103965.1	IL25	PPAPDC3	SPTBN1
AGAP5	KPNA3	QRICH1	SSBP3
BEND7	LINC00964	RCAN2	SSXP10
C1orf86	MALAT1	RP11-1070N10.3	STRN
CAB39L	MLF1	RP11-182J1.16	SYNE2
CBX8	MMP11	RP11-344N10.5	SYPL2
CMTM5	MRPL37	RP11-379F4.7	TFCF
CSPG4P11	MTSS1	RP11-397E7.4	THR8
DDX42	MYBPHL	RP11-724N1.1	UBE3B
DNAH11	MYOZ1	SCN5A	WDR73
EMB	NDST2	SCN10A	ZHX1
GBF1	NEURL	SH3PXD2A	
GORASP1	NPIPA1	SLC2A11	
HCN1	PHLDB2	SMARCB1	

Supplementary Tables 13-14

Information was updated in both tables, and now includes results from the colocalization analysis we performed.

Supplementary note

Pages 37-40:

Supplementary Figure 8 Volcano plots of transcriptome-wide analysis for PR interval duration.

The plots show the results from predicted gene expression analysis in left ventricle (a), right atrial appendage (b), and spleen (c) tissues from GTEx. Analysis was performed with S-PrediXcan using the European meta-analysis summary level results. The x-axis shows the effect size for associations of predicted gene expression and PR interval duration for each gene. The y-axis shows the $-\log_{10}(P)$ for the associations per gene. Each plotted point represents the association results of a single gene. The highlighted genes are significant after Bonferroni correction for all tested genes at the three tissues with a $P < 3.1 \times 10^{-6}$ ($=0.05/(5,977+5,366+4,598)$). Genes with positive effect (blue) showed an association of increased predicted gene expression with PR interval duration. Genes with negative effect (yellow) showed an association of decreased predicted gene expression with PR interval duration.

C

Reviewer #3 (Remarks to the Author):

Dr. Ntalla and colleagues have submitted a manuscript that studies the relationship between genetic variants at a genome-wide scale and the PR interval in a combination of several cohorts and multiple ancestries. The investigators performed a meta analysis of summary data from individual cohort analyses from an additive model adjusted for several demographic and biometric factors as well as principle components to adjust for population stratification. The authors found that there were 149 novel loci associated with the PR interval and that these, in combination with established variants, explain 62.6% of the heritability of PR interval. They further describe the pathophysiologic processes enriched in these PR-related genes.

The PR interval is a fundamental biologic cardiac property that has a relationship to several underlying physiologic functions and disease states. The genomic architecture of the PR interval is therefore of significant interest. The authors are to be commended for such a large international collaboration. The combination of multiple racial/ethnic groups adds to the novelty and appears to have contributed to the discovery many new loci.

I do have several comments and questions that I ask the authors will address and that I hope will strengthen this manuscript.

Major Comments:

1. Rather than use separate discovery and replication cohorts, the authors use a single discovery study design and only considered those with genomewide significance in 60% of the max sample size. The justification for this was based on such a large sample size. Can the authors explain how this is more robust compared to using only 60% of the sample size for discovery and 40% for replication? Are there references to this approach, in particular justifying how the selection of 60% was chosen? Was this 60% of the cohorts, or 60% of the participants? While this approach may be reasonable given such a large overall sample size, note that the non-European groups were relatively small, and also note that there may be a very small number of cohorts contributing most of the non-European participants. Was there equal distribution of non-European ethnic groups in the 60% sample compared to the entire cohort? How many variants reached genomewide significance in the 60% sample, but not in the full analysis?

Authors' reply:

We thank the reviewer for this comment. We would like to clarify that we applied the 60% filtering to the GWAS summary results and this filter was applied in the total number of participants. In other words variants present in <175,831 participants in the GWAS meta-analysis output were not considered for identification of novel loci.

Similarly to our response to the first comment of Reviewer #1 regarding our study design, we decided at the outset to undertake a one-stage big discovery study given our access to a large sample size. To ensure robustness of identified loci we considered for discovery only variants that were present in at least 60% of the maximum sample size ($N_{\text{samples}} \geq 175,831$) in the GWAS summary results. All of our novel loci discovered in the multi-ancestry meta-analysis were discovered in a sample $\geq 212,667$ individuals. Please refer to our response to the first comment of Reviewer #1 and a figure showing the distribution of

the proportion of the total sample size analysed for the 141 novel loci and references to this approach.

2. The number of novel variants is quite large – larger than the number of previously reported variants. Was this a function of just the absolute increase in number of participants included, or a function of combining the different racial/ethnic groups. To get at this question, can the authors tell us how many total novel variants would have been discovered had they only studied European ancestral groups? The authors note that there are 6 novel variants in the European only analyses, but this wasn't entirely clear. Does this mean that 6 novel variants were found in the European only analysis that were not found in the overall meta-analysis, but there were many other novel variants that were found in both the European only and in the combined ancestry analyses. If so, it would be helpful to know how many of the novel variants in all would have been found in the European only analysis, and importantly, how many novel variants were discovered because of the addition of the other racial/ethnic cohorts.

Authors' reply:

In our GWAS multi-ancestry meta-analysis, we included approximately 300,000 individuals, the largest prior study included ~100,000 individuals (PMID: 30046033). Additionally, we considered variants imputed using the 1000 Genomes reference panel providing improved coverage of genetic variation, the earlier GWASs included variants imputed using the HapMap2 reference panel (PMID: 30046033). These two factors primarily contributed to the identification of the large number of novel variants compared to the number of previously reported variants.

In our submitted manuscript we reported a total of 149 lead variants representing 149 novel loci. Of the 149 loci, 119 reached genome-wide significance in both the multi-ancestry and European only meta-analyses. Twenty-two loci reached genome-wide significance only in the multi-ancestry meta-analysis (P-values of these variants in the European meta-analysis were $< 2.9 \times 10^{-6}$). Eight loci reached genome-wide significance in the European meta-analysis only (P-values of these variants in the multi-ancestry meta-analysis were $< 5.4 \times 10^{-7}$). More information on the overlap of loci identified across the GWAS meta-analyses we performed is provided in Supplementary Table 5, where we provide information on all lead novel variants discovered from our meta-analyses, and Supplementary Figure 7.

Please note that in our revised manuscript following the first comment of Reviewer #1 we are making a clear distinction between the multi-ancestry meta-analysis, which is our primary meta-analysis, and ancestry specific meta-analyses, which are secondary.

3. The assumption in combining the different racial/ethnic cohorts is that variants have a homogeneous effect in all racial ethnic groups and that by combining the ancestries, those with marginal significance now become significant. However, since the European group is significantly larger, variants with European-specific significance would remain significant even if the effects are smaller in the non-European cohorts. How many of the variants had heterogeneous effects across ancestries/ethnicities?

Authors' reply:

We performed heterogeneity tests for all novel identified loci for the multi-ancestry meta-analysis across individual studies and across ancestry groups, and there was no evidence of heterogeneity for the newly identified loci.

We have now inserted the heterogeneity P values into Table 1.

Manuscript changes:

Table 1:

Heterogeneity P-values have been inserted into Table.

4. For adjustment of population stratification, were the PCs used cohort-specific or was this standardized across each racial/ethnic group. How many PCs were used in the models, and was this number uniform across all analyses or derived independently by each cohort?

Authors' reply:

The PCs used were cohort specific and were not standardised across each racial/ethnic group. We have now included information on PCs used by each study into Supplementary Table 2.

5. The authors report at least one rare novel variant that was associated with PR interval. Note that the cutoff for genomewide significance is based on common variation (MAF $\sim >1\%$) across the genome in European cohorts. There more common variation in non-European, particularly African, ancestral populations. There is even more variation once the MAF falls $< 1\%$. Can the authors justify their use of this cutoff for the discovery of the rare variant reported?

Authors' reply:

As we have described in response to the first comment of Reviewers #1 and #2, as part of our quality control pipeline we excluded variants with MAC effective < 10 . We then further excluded variants if they were not available in at least 60% of the maximum sample size of the GWAS summary results. This quality control pipeline resulted in us mainly checking results for common variants, and thus we report only one significant association at the *SPSB3* locus. This variant has a frequency of $\sim 1\%$ in the multi-ancestry meta-analysis.

6. Similarly, I do have another concern regarding multiple hypothesis testing. Again, Bonferroni correction is used for multiple-hypothesis testing in one analysis using $\sim 1M$ representative common variants. However, the authors here performed several analysis (all cohorts, European only, African only, Hispanic only, Brazilian only, autosomal, non-autosomal). The opportunity for discovery here is additive, but the Bonferroni threshold for significance did not vary. Granted, most of the novel discovery was made in the overall group and the European only analysis, but by simply looking at different groups adds to the risk of false discovery. Do the authors have any justification for not adjusting further the multiple hypothesis threshold?

Authors' reply:

Our primary analysis was the multi-ancestry GWAS meta-analysis and in the revised manuscript we now report these findings as our main results. We did perform ancestry specific and X chromosome analysis and now report these findings as secondary analyses in the paper. In our response to a similar query to Reviewer #1 we provide detailed information on our new reporting of results.

7. I'm not sure the polygenic risk score here adds significantly. Note that the PR interval itself is not a pathogenic entity, rather itself an endophenotype. While interesting that the PGRS is associated with a few cardiovascular diseases, what is the added value of the PGRS on top of the PR interval, which itself is easy to measure. Is there any added value? Also, can the authors explain why they believe the PGRS was inversely related to AF, while the PR is typically positively related to AF?

Authors' reply:

Epidemiologic data support associations between PR interval and various cardiovascular phenotypes (as we reference in the paper). Use of polygenic risk scores in our study specifically highlights the fact that genetic predisposition to PR interval duration is associated with cardiovascular conditions. Furthermore, we indicate our results provide two further insights. First, we report both the direction and magnitude of effect for polygenic predisposition to PR interval with each cardiovascular trait, which is of potential value for risk prediction (although this is not the focus of the present manuscript). Second, although our findings highlight the aggregate effect of polygenic predisposition, they are complemented by the recognition that such scores may have heterogeneous pathway specific effects. For example, we highlight the relations between specific PR interval associated variants and AF associated variants, as outlined below. In summary, although PR interval itself is not a pathogenic entity and is easy to measure, clarifying the genetic determinants of the PR interval may one day facilitate risk prediction and also better elucidate complex biological pathways between atrioventricular conduction and specific cardiovascular conditions.

Manuscript changes:

Main text:

Page 21:

For example, our findings are consistent with previous epidemiologic data supporting a U-shaped relationship between PR interval duration and AF risk¹. Although aggregate genetic predisposition to PR interval prolongation is associated with reduced AF risk, lead PR interval prolonging alleles are associated with decreased AF risk (e.g., localized to the *SCN5A/SCN10A* locus; **Supplementary Fig. 11**) whereas others are associated with increased AF risk (e.g., localized to the *TTN* locus; **Supplementary Fig. 11**), consistent with prior reports⁸.

Pages 44-45:

Supplementary note:

Supplementary Figure 11 Associations between lead PR interval variants (205 single nucleotide polymorphisms) with atrial fibrillation (AF) risk from recently published AF GWASs.

The X axis refers to $-\log_{10}$ P-value for PR interval. Y axis refers to $-\log_{10}$ P-value for AF risk. Red/orange color indicates the same direction of effect for PR interval and

AF risk, while blue colour indicates the opposite direction of effect for PR interval and AF risk. The different color scheme shows different odds ratio of AF risk. The nearest gene names are labelled if variants were genome-wide significantly associated with both traits. Upper panel (a) is the look-up results from Roselli et al.², and lower panel (b) is the look-up results from Nielsen et al.²⁹

Minor Comments

1. In regards to the statement that there are several monogenic-linked genes that were found in/near the loci associated with PR prolongation, I would point out that the many of these genes in

actuality have weak evidence for their monogenic roles, except for HCN4 and RYR2 as the authors point out.

Authors' reply:

We have gone back and reviewed the information on all candidate genes at the novel loci. Following your suggestion we have been more stringent in our reporting of monogenic genes and now only highlight in the paper candidate genes at 8 loci with strong support as being genes causing either inherited arrhythmic syndromes or cardiomyopathies. We have also made edits to Supplementary Tables 18 and 19 so presentation of data we hope is clearer.

Manuscript changes:

Main text

Page 20

Common variants in/near genes associated with **monogenic** arrhythmia syndromes were observed, suggesting these genes **may also** affect atrioventricular conduction and cardiovascular pathology in the general population. Apart from *DSP*, *DES* and *GJA5* **discussed** above, our analyses indicate **2** additional candidate genes (*HCN4* and *RYR2*). *HCN4* encodes a component of the hyperpolarization-activated cyclic nucleotide-gated potassium channel which specifies the sinoatrial pacemaker “funny” current, and is implicated in sinus node dysfunction, AF, and left ventricular noncompaction³⁴⁻³⁶. *RYR2* encodes a calcium channel component in the cardiac sarcoplasmic reticulum and is implicated in catecholaminergic polymorphic ventricular tachycardia³⁷.

2. The authors point out that there were a number of novel missense variants, implying that these were the causative variants. While this is highly likely, I think it is still important to point out that these variants are still only a representation of multiple variants in high LD and that the causative variant(s) may not be those particular missense variants.

Authors' reply:

We describe the properties of lead variants and this includes the number of novel missense variants at both novel and previously reported loci. We highlighted and briefly discussed two of these, the rare missense variants in *SPSB3* and *MYH6*. We do not wish to imply the missense variants were the causal variants at these loci. To ensure there is no misinterpretation in our reporting we have edited the paragraph describing these results.

Manuscript changes:

Pages 16-17:

The majority of the lead variants at the **149** novel loci were common (minor allele frequency, MAF>5%). We observed 6 low-frequency (MAF 1-5%) variants, and one rare (MAF<1%) predicted damaging missense variant. **The rare variant** (rs35816944, p.Ser171Leu) **is** in *SPSB3* encoding SplA/Ryanodine Receptor Domain and SOCS Box-containing 3. *SPSB3* is involved in degradation of **the transcription factor** SNAIL, which regulates the epithelial-mesenchymal transition¹⁵, and has not been previously associated with cardiovascular traits. **At *MYH6*, a previously described locus for PR interval^{6,10}, sick sinus syndrome¹⁶, AF and other cardiovascular traits¹⁷, we observed a novel predicted damaging missense variant in *MYH6* (rs28711516, p.Gly56Arg). *MYH6* encodes the α -heavy chain subunit of cardiac myosin.** In total,

we identified missense variants in genes at 11 novel, 1 from the European subset meta-analysis, and 6 previously reported loci (Supplementary Table 12). These variants are a representation of multiple variants at each locus which are in high LD and thus may not be the causative variant.

3. The authors explain there were several exclusion criteria, including use of certain medications. Was medication data reliably available in all cohorts? If not available, was the assumption that the participants were not on these medications. For exclusion criteria, did the authors mean those with paced rhythms were excluded, or those with pacemakers? Subtle but important distinction. The latter would be per participant history which may not be available from all cohorts.

Authors' reply:

Not all studies had the information to apply all the exclusions we requested, and one of these was the medication data. Thus, the assumption we have taken is that participants were not on the medications. We had requested individuals with a pacemaker to be excluded, where this information was available this was applied, but again not all studies had this information. We have made some edits to the methods to make it clear not all studies could apply the exclusions we had requested.

Manuscript changes:

Online Methods

Page 23:

PR interval phenotype and exclusions

The PR interval was measured in milliseconds from standard 12-lead electrocardiograms (ECGs), except in the UK-Biobank in which it was obtained from 4-lead ECGs (CAM-USB 6.5, Cardiosoft v6.51) recorded during a 15 second rest period prior to an exercise test (Supplementary Note). We requested exclusion of individuals with extreme PR interval values (<80ms or >320ms), second/third degree heart block, AF on the ECG, or a history of myocardial infarction or heart failure, Wolff-Parkinson-White syndrome, those who had a pacemaker, individuals receiving class I and class III antiarrhythmic medications, digoxin, and pregnancy. Where data was available these exclusions were applied.

4. The transcriptome analysis though interesting, does not add significantly to the primary findings. Most salient, the findings of the transcripts not present in the primary analysis are not replicated and their importance should be minimized.

Authors' reply:

We appreciate the reviewer's comment and have modified the language in the manuscript to deemphasize the analysis.

Manuscript changes:

Main text

Page 17:

In an exploratory analysis, we also performed a transcriptome-wide analysis to evaluate associations between PR interval duration and predicted gene expression in LV and RAA. We identified 113 genes meeting our significance threshold ($P < 3.1 \times 10^{-6}$, after Bonferroni correction), of which 91 were localised at PR interval loci (within 500kb from a lead variant; **Supplementary Table 14, Supplementary Fig. 8**).

5. Why did the authors put both height and BMI in the primary models? Along these lines, why not (height + weight) or (height + weight + BMI)?

Authors' reply:

Height and BMI (and weight) are associated with cardiac conduction ECG phenotypes (but not repolarization). In our analysis we wished to study electrophysiology beyond morphology due to body size, hence covariates selected. It is of course possible that our analysis may have missed variants that influence PR interval through height.

Reviewers' Comments:

Reviewer #1:

Remarks to the Author:

The authors have responded to the reviewer's comments satisfactorily and have revised their manuscript appropriately.

Norihiro Kato

Reviewer #2:

Remarks to the Author:

Authors have responded to my request for negative control tissues to be included in their bioinformatic follow-up. The COLOC analysis is also a valuable addition to the manuscript. I also think that revising the PRS section to focus only on European data is more scientifically valid considering the current issues with applying these across ancestry groups with the existing resources available.

I think the authors have adequately responded to my comments on the original manuscript and as far as I can tell, have done the same for other reviewer comments. I am happy for this to be accepted for publication as long as the other reviewers think their comments/concerns have been addressed to their satisfaction.

Reviewer #3:

Remarks to the Author:

I have reviewed the authors' reply to my queries and am satisfied with the responses. Making a distinction between the primary analyses were and the secondary/exploratory analyses was important. I think it is a fair assumption that this is what the authors had in mind when starting their analyses. I also thank the authors for further clarifying the contribution of novel loci from the European versus the combined cohort.